



# Understanding nighttime methane signals at the Amazon Tall Tower Observatory (ATTO)

Santiago Botía B.[1], Christoph Gerbig[1], Julia Marshall[1], Jost V. Lavric[1], David Walter[1,2],
Christopher Pölhker[2], Bruna Holanda[2], Gilberto Fisch[3], Alessandro Carioca de Araújo[4], Marta O. Sá[5],
Paulo R. Teixeira[5], Angélica F. Resende[5], Cleo Q. Dias-Junior[6], Hella van Asperen[7], Pablo S. Oliveira[8],
Michel Stefanello[8], and Otávio C. Acevedo[8]

[1]Max Planck Institute for Biogeochemistry, Hans-Knoell-Straße 10, 07745 Jena, Germany
[2]Max Planck Institute for Chemistry, 55020 Mainz, Germany
[3]Centro Técnico Aeroespacial, São José dos Campos, SP, Brazil
[4]Empresa Brasileira de Pesquisa Agropecuária (EMBRAPA), Trav. Dr. Enéas Pinheiro, Belém, PA, Brazil
[5]Instituto Nacional de Pesquisas da Amazônia (INPA), Av. André Araújo 2936, Manaus, AM, Brazil
[6]Instituto Federal de Educação, Ciência e Tecnologia do Pará, Belém, PA, Brazil
[7]Institute of Environmental Physics, University of Bremen, Otto-Hahn-Alle 1, 28359, Bremen, Germany
[8]Departamento de Física, Universidade Federal de Santa Maria, Av. Roraima 1000, Santa Maria, RS, Brazil

**Correspondence:** Santiago Botía B. (sbotia@bgc-jena.mpg.de)

**Abstract.** Methane ($CH_4$) atmospheric mixing ratio measurements are analyzed for the period between June 2013 and November 2018 at the Amazon Tall Tower Observatory (ATTO). We describe the seasonal and diurnal patterns of nighttime events in which $CH_4$ mixing ratios at the uppermost (79 m a.g.l) inlet are significantly higher than the lowermost inlet (4 m a.g.l) by 8 ppb or more. These nighttime events were found to be associated with a wind direction originating from the southeast and wind speeds between 2 and 5 m s$^{-1}$. We found that these events happen under specific nighttime atmospheric conditions when compared to other nights, exhibiting less variable sensible heat flux, low net radiation and a strong thermal stratification above the canopy were found. Our analysis indicates that even at wind speeds of 5.8 m s$^{-1}$ the turbulence intensity, given by the standard deviation of the vertical velocity, is suppressed to values lower than 0.3 m s$^{-1}$. Given these findings, we suggest that these nighttime $CH_4$ enhancements are advected from their source location by horizontal non-turbulent motions. The most likely source location is the Uatumã River, possibly influenced by dead stands of flooded forest trees that may be enhancing $CH_4$ emissions from those areas. Finally, biomass burning and the Amazon River were discarded as potential $CH_4$ sources.

## 1 Introduction

Atmospheric methane ($CH_4$) is, after carbon dioxide ($CO_2$), the most important anthropogenic greenhouse gas (Ciais et al., 2013). In the last decades atmospheric $CH_4$ has followed contrasting trends. Following decades of growth, during the period between 1999 to 2006, atmospheric $CH_4$ mixing ratios were stable (Dlugokencky E. et al., 2011), whereas during the last decade a steep growth has been reported (Nisbet et al., 2016). Currently the reasons for these trends are not clear and several explanations have been proposed (Nisbet et al., 2016; Schaefer et al., 2016; Turner et al., 2017; Rigby et al., 2017; Howarth, 2019). What is evident in the current debate is that $CH_4$ emissions from tropical wetlands are the single largest source of





uncertainty to the global $CH_4$ budget (Kirschke et al., 2013; Saunois et al., 2016). Recent estimates suggest that Amazon $CH_4$ emissions could contribute about a third of the global wetland $CH_4$ emissions (Pangala et al., 2017). Therefore, better knowledge of the seasonal and diurnal dynamics of these emissions will provide valuable insights for developing process-based models that more accurately represent $CH_4$ emission and uptake (Turner et al., 2019). Our long term measurements of $CH_4$

mixing ratios at the Amazon Tall Tower Observatory (ATTO) provide an opportunity for better understanding such seasonal and diurnal dynamics, which are necessarily linked to atmospheric transport (Gloor et al., 2001).

    $CH_4$ is transported from surface sources, which in the Amazon are dominated by wetland ecosystems (Saunois et al., 2016; Wilson et al., 2016), to the upper levels of the troposphere. Therefore, the vertical profile of $CH_4$ mixing ratios in the troposphere generally decreases with height, showing higher mixing ratios in the boundary layer (Miller et al., 2007; Beck et al.,

2012; Webb et al., 2016). The profiles discussed in previous studies have been based on airborne measurements taken at daytime under well-mixed conditions during specific campaigns (e.g., Beck et al., 2012) or during regular sampling programs with aircraft flights at least twice per month (e.g., Gatti et al., 2014; Webb et al., 2016). These studies provide valuable insights about regional atmospheric transport (Miller et al., 2007), spatial distribution of dominant $CH_4$ sources (Wilson et al., 2016) and the seasonal cycle across the Amazon region (Webb et al., 2016).

Closer to the surface, the proximity to the canopy and to the $CH_4$ sources becomes more important, dominating the variability of $CH_4$ mixing ratios at diurnal time scales. This strong variability is driven by a heterogeneous spatial distribution of sources (Gloor et al., 2001) and the complexity of atmospheric transport mechanisms in the first meters of the boundary layer (Stull, 1988). Despite these complexities, mixing ratio measurements close to the surface are the most feasible way to perform long-term measurements at high temporal resolution that can provide insights at diurnal and seasonal time scales. However, such

type of measurements are sparse in this region, particularly for $CH_4$.

    There are two studies that provide an idea of the diurnal variability of $CH_4$ mixing ratios in the Amazon region (Carmo et al., 2006; Querino et al., 2011). Both of these studies were conducted in upland forest sites, similar to the present study. Carmo et al. (2006) performed $CH_4$ mixing ratios measurements during the dry and wet season at three different sites in the Amazon. At all sites they focused their profile measurements on the canopy layer, which included one sampling inlet not more

than 10 m above the canopy top. This study found that mixing ratios were generally higher at nighttime than during the day at all sites. As Carmo et al. (2006) focused on identifying local sources and calculating a local budget, the authors do not specify the amplitude of the diurnal cycle for the dry and wet seasons. Querino et al. (2011) provide one of the few records showing the diurnal variability of $CH_4$ above the canopy (53 m), as well as mixing ratio profiles within the canopy. $CH_4$ mixing ratios above the forest were higher during the night than during the day. The amplitude of the diurnal cycle was particularly large

in July (> 30 ppb), whereas in November there was no diurnal variability (Querino et al., 2011). Vertical $CH_4$ profiles inside the canopy were found to decrease with height, during both day and night, but more strongly during nighttime measurements, agreeing with Carmo et al. (2006). Both Querino et al. (2011) and Carmo et al. (2006) attributed this feature to a source located in the soil, and they both concluded that the upland forest is a source of $CH_4$. The observed high nighttime $CH_4$ mixing ratios were explained by referring to a shallower nocturnal boundary layer that led to accumulation of $CH_4$: above the canopy (at 53

m) in Querino et al. (2011) and throughout the canopy layer in Carmo et al. (2006).





In the present study we describe similar nighttime $CH_4$ mixing ratio patterns using the unprecedented long-term measurements at ATTO. This unique data set allows us to better describe the seasonal and diurnal variability of nighttime $CH_4$ signals at the 80 m tower. Furthermore, we provide a detailed analysis of the dominant atmospheric conditions under which high $CH_4$ mixing ratios are measured during the night at the top of the tower. Finally, we suggest possible sources of this nighttime $CH_4$

together with a description of the transport mechanisms that could be responsible for the vertical and horizontal transport of $CH_4$ in the nocturnal boundary layer.

## 2 Data and Methods

### 2.1 Site description

The Amazon Tall Tower Observatory (ATTO) research station was described in detail by Andreae et al. (2015); here we will

only highlight aspects relevant to the current study. The ATTO site is located in the Uatumã Sustainable Development Reserve (USDR), which is 150 km northeast of the city of Manaus, in Central Amazonia. The site was built on a plateau (130 m-a.s.l) which is surrounded by a large drainage network composed of depressions or valleys at lower elevation, that together with the plateaus form a small-scale heterogeneous topography with maximum height gradients of about 100 m (see Fig. 1).

The site is located in the interfluve between the Uatumã River and its tributary, the Abacate River. Both of these rivers flow

in a NW to SE direction, merging their waters further southeast of the site. The flow of the Uatumã River is controlled by the Balbina reservoir, a hydroelectric dam located 55-60 km to the northwest of ATTO. The natural flood pulse of the Uatumã River was disturbed by the Balbina reservoir, causing large tree mortality along the riverside (Assahira et al., 2017; Resende et al., 2019). It is worth noting that the Abacate River has not suffered hydrological disturbances, still presenting a natural flooding seasonality.

The vegetation of the USDR is composed of different ecosystems. Upland dense forest (*terra firme*) is the characteristic vegetation on the plateaus and slopes, with the highest canopy when compared to the surrounding valleys (Andreae et al., 2015). The canopy height at ATTO is around 37 m, but the mean canopy height over the plateau is $20.7 \pm 0.4$ m (Andreae et al., 2015). In between the permanent flowing river channels and the *terra firme* forest, a savanna-type ecosystem on white-sand soil (*campina*) and forest on white-sand soil (*campinarana*) are found. In the *campina* there are smaller shrubs and trees with a

vegetation height of $13.1 \pm 2$ m. The *campinarana* has a canopy with emergent trees up until 17 m (Klein and Piedade, 2019). Along the Uatumã and the Abacate rivers, seasonally flooded black-water forest (*igapó*) is the dominant type of vegetation (Andreae et al., 2015).

The regional atmospheric circulation has a seasonal pattern driven by the annual north-south shift of the Inter-tropical Convergence Zone (ITCZ) over the Atlantic Ocean (Andreae et al., 2012, 2015; Pöhlker et al., 2019). During the dry season,

which we define here from July to October, ATTO is located south of the ITCZ. During the wet season, defined here from February to May, ATTO is located north of the ITCZ. This seasonal shift influences the origin of the air masses entering the continent and arriving at ATTO. During the dry season the prevailing wind direction is from the east, whereas the wind direction during the wet season is predominantly from the northeast (Andreae et al., 2015). Easterly winds during the dry season transport





air masses containing signals from biomass burning, occurring mainly in the eastern part of the arc of deforestation in Brazil. The northeasterly winds during the wet season are transported over a large fetch of continuous rainforest, yet containing high background mixing ratios of greenhouse gases due to the influence of the northern hemisphere (Andreae et al., 2012, 2015). The dry and wet seasons were defined using precipitation data collected at the site with 30-minute resolution covering a period

from January 2013 to December 2018. The highest precipitation values are recorded from February to May, with a maximum of 300 mm month$^{-1}$ during March, while the lowest values, at below 100 mm month$^{-1}$, are found for the months between July and October (Fig. 2).

## 2.2   CH$_4$ mixing ratio sampling system

Our continuous measurements system was installed in March 2012 in the 80-m-tall tower. This tower was built during the

pilot phase of an ongoing measurement program. There are five air inlets located at 79, 53, 38, 24 and 4 m above ground. At these heights, we measure atmospheric mixing ratios of CH$_4$, carbon dioxide (CO$_2$) and carbon monoxide (CO) using two Picarro Analyzers, the G1301 for CH$_4$ and CO$_2$ and the G1302 for CO$_2$ and CO. The G1301 analyzer (Serial CFADS-109) provides data with a standard deviation of the raw data below 0.05 ppm for CO$_2$ and 0.5 ppb for CH$_4$. For the G1302 (Serial CKADS-018), the standard deviation of the raw data is 0.04 ppm for CO$_2$ and 7 ppb for CO. For more details on precision

and long-term drift see Andreae et al. (2015). Similar to Winderlich et al. (2010), downstream of each sampling line 8 l stainless steel spheres act as buffer volumes. This buffer system provides ideal air mixing characteristics to have continuous near-concurrent measurements at all heights (Winderlich et al., 2010). In Winderlich et al. (2010) the buffers integrate the air signal from every inlet with an e-folding time of approximately 37 min, with a flow of 150 standard cubic centimeter per minute (sccm), at 700 mbar. As we have two analyzers our time resolution is higher with an e-folding time of approximately 18 min

(8 l/(150x2) sccm at 700 mbar). Therefore, we assume that the 15 min averages used in Sect. 3.2.3 correspond to independent samples. The standard data available at *http://attodata.org* are at 30-minute resolution.

## 2.3   Meteorological instrumentation

For this study we use wind direction, wind speed, air temperature, net radiation, precipitation and soil moisture. Sonic anemometers are fixed at 14, 22, 41, 55 and 81 m (above ground level; a.g.l.), but in this study we mainly use wind speed

and direction at 81 m (Windmaster, Gill Instruments Limited). In Sect. 3.1.1 we show wind profiles for specific nights, for these we use additional data from the sonic anemometers (model CSAT3, Campbell Scientific, Inc.) at 14, 41 and 55 m which perform fast response wind (u, v, and w) measurements. Air temperature is measured at 10 heights: 81, 73, 55, 40, 36, 26, 12, 4, 1.5 and 0.4 m (a.g.l) with a Termohygrometer (C215, Rototronic Measurement Solutions, UK.). Net radiation is measured with a Net radiometer (NR-Lite2, Kipp and Zonen, Netherlands) at 75 m (a.g.l). For precipitation data, a rain gauge (TB4,

Hydrological Services Pty. Ltd., Australia) is installed at 81 m, and for soil moisture a water content reflectometer (CS615, Campbell Scientific Inc., USA) provides data for the depths: 0.1, 0.2, 0.3 0.4, 0.6 and 1 m.





## 2.4 Time period of data used

In the present study we have used $CH_4$ mixing ratio and meteorological data at different time resolutions. When mentioning meteorological data, we refer to the variables described in 2.3. To provide more clarity we specify what type of data were used in each section. In Sect. 3.1, 3.2.1 and 3.2.2 we used $CH_4$ mixing ratios and meteorological variables at 30-minute resolution. The $CH_4$ mixing ratio record covers the period from June 2013 to November 2018, which enabled us to study the diurnal and seasonal variability within this period. In Sect. 3.2.3, we used high-frequency (10 Hz) meteorological data, in particular all wind components (u, v and w), in order to associate turbulence regimes with $CH_4$ mixing ratios at 15-minute resolution. More on the assumptions to link high-frequency wind data with 15-minute mixing ratios is given in Sect. 3.2.3. In Sect. 3.3 we use CH4, CO and black carbon (BC) concentrations, all at 30-minute, to assess the influence of biomass burning emissions in our CH4 signals. The BC data were obtained using the Multiangle Absorption Photometer (MAAP, model 5012, Thermo Electron Group, Waltham, USA), as described in detail in Saturno et al. (2018). In Table 1, we provide a list of the data used in each section, specifying the time resolution and the period of time used.

## 2.5 $CH_4$ gradient definition

We have defined a $CH_4$ gradient as the $CH_{4_{grad}} = CH_{4_{79m}} - CH_{4_{4m}}$. We refer to a positive gradient when $CH_{4_{grad}} > 0$, or to a negative gradient when $CH_{4_{grad}} < 0$. Note that positive gradients are related to higher $CH_4$ mixing ratios at 79 m than at 4 m, while negative gradients to higher mixing ratios at 4 m. Throughout this paper we also use a 8 ppb threshold for classifying positive gradients and negative gradients. In Sect 3.1 we use three classes. The first one refers to very strong positive gradients ($CH_{4_{grad}} > 8$ ppb); the second one to gradients in between -8 ppb and 8 ppb (-8 < $CH_{4_{grad}} < 8$ ppb); the third one to very strong negative gradients ($CH_{4_{grad}} < -8$ ppb). In Sect. 3.2, we have limited our analysis to two classes, $CH_{4_{grad}} > 8$ ppb, and $CH_{4_{grad}} < 8$ ppb. The motivation to use 8 ppb as the threshold value is to leave out small mixing ratio variations and select very strong events. The $\pm$ 8 ppb threshold is conservative and filters for strong gradients, if we consider that the annual global increase in atmospheric $CH_4$ during the last three years was 7.06, 6.95 and 10.77 ppb yr$^{-1}$ for 2016, 2017 and 2018 respectively (Dlugokencky and NOAA). It is always stated in the text which of these classes is being considered.

## 3 Results and Discussion

## 3.1 Seasonal and diurnal patterns of $CH_4$ gradients at ATTO

In this section, we focus on the gradient between the $CH_4$ mixing ratios measured at 79 m (above the canopy) and at 4 m (within the canopy). A strong positive gradient, when $CH_{4_{grad}} > 8$ ppb (i.e. $CH_4$ mixing ratio at 79 m is higher than at 4 m) could be associated with an air mass measured at 79 m that is disconnected from the air within the canopy, resulting from atmospheric transport of *non-local* sources, with *non-local* referring to areas different from the plateau where the tower is located. For very strong negative gradients, when $CH_{4_{grad}} < -8$ ppb (i.e. $CH_4$ mixing ratio at 79 m is lower than at 4 m), we assume they could





be associated with local sources within the canopy. Later in Sect. 3.3 we discuss what are the potential *non-local* sources that might be driving the positive $CH_4$ gradients, hereafter simply referred to as positive gradients.

In Fig. 3 we show a monthly time series of the $CH_4$ gradient at ATTO. The data indicate that for all years except 2016, monthly mean positive gradients occur mainly during the dry season, whereas the mean gradient is close to zero during the wet

season. The monthly mean gradient is significantly different from zero during the dry months, with p-values (two sided t-test) lower than 0.01. This positive gradient at ATTO is more pronounced during the month of August, when the lowest precipitation values are less than 50 mm (Fig. 2). A different behavior is observed for 2016, in which a strong negative gradient during June and July suggests a possible $CH_4$ source within the canopy measured at 4 m. The reason for this is unclear, yet we provide some ideas discussed later in this section. The dry season peak is also seen at the other measurement heights (not shown here),

but at 79 m the measured $CH_4$ mixing ratios are the highest, indicating that non-local sources are predominant during this period of time.

It is interesting to note that in the dry season also the largest variability in the gradient was observed, as opposed to the reduced variability during the wet season and transition months such as December and January. The difference in variability can be partly explained by the seasonal shift of the air masses arriving at ATTO, which was described in Sect.2.1. During

the dry season, easterly winds transport strong signals of trace gases and aerosols associated with biomass burning activity in the southern and southeastern region of the Amazon basin (Andreae et al., 2015; Saturno et al., 2018). During the wet season relatively clean air arrives from the northeast, travelling along a very large fetch of rainforest (Andreae et al., 2015). However, as we will show in Sect. 3.3.3, biomass burning alone cannot explain the positive gradient events at ATTO.

The positive gradients observed during the dry season are associated with nighttime $CH_4$ mixing ratio peaks at 79 m. When

separating the measurements into daytime and nighttime, and grouping the half-hourly averaged gradients calculated from the measurements into three classes ($CH_{4grad} > 8$ ppb; $-8 < CH_{4grad} < 8$ ppb; $CH_{4grad} < -8$ ppb), one can estimate the fraction of time (separately for night and day) corresponding to each class. We hypothesize that the $> 8$ ppb class is the result of $CH_4$-rich air brought to ATTO and measured at 79 m, whereas the $< -8$ ppb class results from a nearby source within the canopy measured at 4 m. Figure 4 depicts the fraction of time for daytime and nighttime measurements for each of the classes described above.

For all months, daytime measurements (Fig. 4a) are within the -8 to 8 ppb range over 60% of the time. Gradients lower than -8 ppb or higher than 8 ppb are more frequent and increase their contribution to the total time during May, June, July, August and September. For these months negative gradients are measured 9%, 19%, 22%, and 18% and 9% of the total daytime hours, while positive gradients are found 5%, 7%, 12%, 14% and 10%.

Nighttime measurements, in general, show a larger contribution of positive gradients to the total time (Fig. 4b). Interestingly,

nighttime positive gradients occur in all months of the year, from 4.1% of the time in January to 43% of the time in August. The highest percentages are recorded during the dry season months of July (30%), August (43%) and September (30%). Unsurprisingly, these months also have the highest mean nighttime gradients, with $2.6 \pm 23$ ppb, $9.7 \pm 21$ ppb and $6.2 \pm 23$ ppb for July, August and September, respectively. Moreover, August has the highest nighttime median gradient (5.1 ppb), providing further evidence that positive gradients are more frequent during nighttime in this month. The maximum nighttime

positive gradients were observed between 03:00 and 06:00 am local time (not shown), with values larger than 130 ppb. During





daytime, the maximum positive gradients occur between 06:00 and 08:00, with values over 150 ppb. Note that these gradients are generated during nighttime and can persist for a couple of hours until the erosion of the nocturnal boundary layer and subsequent growth of the convective boundary layer, which occurs between 08:00 and 09:00 am (local time) (Fisch et al., 2004; Carneiro, 2018).

The mean diurnal cycles of the $CH_4$ gradient measured between 79 m and 4 m (hereafter mean diurnal cycle gradient) for dry and wet seasons provide interesting insights. The amplitude of the mean diurnal cycle gradient during the dry season (12.1 ppb) is 4 times larger than that of the wet season (2.7 ppb). This substantial difference can be attributed to two main reasons. First, due to strong nighttime positive gradients during the dry season, the maximum mean (7.1 ppb) is much larger than the maximum mean of the wet season (0.5 ppb). Interestingly, both of these mean maxima occur at night, indicating that nighttime

positive gradients are pulling up the mean in both seasons. Second, during the dry season the daytime planetary boundary layer is higher by a few hundred meters as a result of a larger sensible heat flux caused by the higher incoming shortwave radiation during this season (Fisch et al., 2004; Carneiro, 2018). This directly affects daytime $CH_4$ mixing ratios, because $CH_4$ molecules will be diluted in a larger volume, leading to lower $CH_4$ mixing ratios at 79 m. This boundary layer effect together with higher $CH_4$ mixing ratios at 4 m compared to 79 m during the dry season yields a lower dry season daytime mean minimum of -5.0

ppb, whereas the mean minimum during the wet season is -2.2 ppb. Another possibility that might contribute to this seasonal difference is local production of $CH_4$ during the wet season. Though we lack long-term $CH_4$ flux measurements at the site, we can infer a potential local source during the wet season considering that the mean monthly gradient during daytime hours of the wet season is always negative (not shown), meaning that the $CH_4$ mixing ratio at 4 m is higher than at 79 m. Although outside of the scope of this study, strong negative gradients are more common during daytime, reaching differences as large

as -455 ppb, measured in May of 2014. In contrast, the largest negative nighttime gradient measured is -236 ppb, occurring in October of 2015. The 4 m inlet is too high above the soil to assume that this $CH_4$ signal comes directly from the soil below, but is well within the canopy, indicating that the source must be local possibly within a horizontal distance of few hundred meters. The event in May 2014, coincided with a strong signal measured for carbon monoxide (CO) with the same timing, but not for carbon dioxide ($CO_2$), which suggests a source not related to combustion.

The mechanism producing this strong $CH_4$ signal within the canopy is currently under investigation, yet here we discuss what the potential sources could be. Our first thought for the strong negative gradient during daytime is that the soil on the plateau is producing $CH_4$ episodically. Given some additional parameters, we can calculate the water-filled pore space (WFPS) for the depth (60 cm) of maximum soil moisture content of 0.35 $m^3 \, m^{-3}$. Based on Andreae et al. (2015) we know that 85% of the plateau is clay, and thus use a soil particle density of 2.86 $g \, cm^{-3}$ (Schjønning et al., 2017)). Also from Andreae et al.

(2015) we use a bulk density of 1.1 $g \, cm^{-3}$. This results in a WFPS of 57% which could enhance the abundance of anaerobic micro-sites where $CH_4$ can be produced. Upland *terra-firme* soils are generally considered as $CH_4$ sinks (Dörr et al., 1993; Dutaur and Verchot, 2007; Saunois et al., 2016), but at local scales the soil can become a source depending on the balance between $CH_4$ production and oxidation (Verchot et al., 2000). Moreover, tree stems were found to play an important role as conduits for soil-generated $CH_4$ in upland *terra-firme* tropical forest (Welch et al., 2019). These findings, together with our

data, suggest that the upland *terra-firme* $CH_4$ sink at ATTO need to be further studied.





A second option is related to daytime anabatic flows within the canopy transported from the depressions to the plateau (Tóta et al., 2012). These anabatic flows could transport $CH_4$ produced in saturated soils of these low-lying areas. Recent (May 2019) unpublished $CH_4$ mixing ratio measurements suggest a possible source at the lowest point of one depression close to the ATTO site. A further possibility, not likely explaining the complete signal at 4 m, but probably contributing to it, might be termite

production within the canopy. Termite $CH_4$ production is common in tropical ecosystems (Sanderson, 1996), however flux measurements from this source have not been performed at the site. The 2016 episode mentioned earlier provides evidence that $CH_4$ mixing ratios at 4 m could be strongly enhanced by unknown processes that need further research. The natural variability of upland *terra-firme* $CH_4$ sources or sinks could have been altered by the El Niño event that began in 2015 and lasted until early 2016, as shown by Pfannerstill et al. (2018) for OH at the ATTO site. For these episodes, which occurred during June and

July (see Fig. 3), the enhancement at 4 m lasted for more than 5 hours presenting the onset at nighttime and maintained during daytime. Furthermore, the air within the 4 m layer above the ground seemed to be strongly decoupled from the layers above as none of the upper inlets measured the signal observed at 4 m (not shown). Therefore, the 4 m episodes described here are very likely a combination of an enhanced source and the stability conditions of the air within the canopy for those specific dates.

### 3.1.1 Example of $CH_4$ gradient events

To give the reader an idea of how strong these nighttime positive gradients can be compared to nights with no gradient, three case studies are presented in Fig. 6. In addition to the $CH_4$ time series measured at different levels, profiles of virtual potential temperature and mean wind speed are shown to assess the air coupling above and within the canopy.

In Fig. 6a we show a night in which a very strong positive gradient occurred. $CH_4$ mixing ratios at all heights follow the same trend for the first hours of the night. From 19:00 to 00:00 there is a common increase of about 15 ppb at all heights. At

22:00 the $\Theta_v$ above the canopy shows a mild gradient with the canopy slightly cooler than the layers above. The $\Theta_v$ in the layer between 40 m and 80 m is constant, and the wind speed profile between the same heights changes by about 1 m s$^{-1}$. After 00:00 and before 01:00, there is a sudden and abrupt divergence of $CH_4$ mixing ratios measured at 79 m. This increase in $CH_4$ is not seen at the first three measurement levels, 4 m, 24 m and 38 m, while at 53 m is observed with some delay and with less intensity. The divergence at 79 m reaches a $CH_4$ peak of 1960 ppb at around 04:00, at which point the lowest three inlets show

$CH_4$ mixing ratios lower than 1880 ppb. Note that at this time there is a strong thermal inversion for the air above the canopy and the wind speed at 81 m decreases to almost 1 m s$^{-1}$. After sunrise, as the canopy and the first meters of the boundary layer are heated by incoming radiation, mixing ratios at all heights converge to the same behavior and show no significant differences. Note that the $\Theta_v$ profile at 08:00 decreases slightly with height and the wind speed has increased to more than 3 m s$^{-1}$ at 81 m. The duration of this positive gradient, considering the time in which the 79 m inlet had mixing ratios higher

than 1880 ppb, was about 5 hours. These positive gradient events are very common in our time series and, as we said before, are more frequent during August, but also occur in July and September. Positive gradient events vary mainly in their duration and the magnitude. For the case shown in Fig. 6a, we can see that the decoupling between the air above and within the canopy was very strong up until 53 m. At the 53 m level the signal, first measured at 79 m, arrived about 30 minutes later when the $CH_4$ mixing ratio began to increase, but at the lower levels the behavior is completely independent. The decoupling is due to





a very strong thermal inversion that obstructs vertical mixing, which could be triggered by wind shear under stable conditions (Mahrt, 2009).

For other cases in which we observed positive gradient events during nighttime, the signals measured at 79 m are then measured at the lowest inlet (4 m) after some time, varying from 30 minutes to 1.5 hours (see Fig. 6b). For these situations,
when the coherent response at the lowest levels is within the next half an hour, it could potentially be explained by the sequential sampling (top-bottom) and the buffer volume in our measurement system. However, in the case of such a subsequent top-down signal we also find that the decoupling between the air above and within the canopy is weaker or non-existent.

The $\Theta_v$ profiles above the canopy at 22:00 and 02:00 show a very similar gradient with a weak thermal inversion. Interestingly, at 02:00 when the $CH_4$ mixing ratio at 79 m reached the maximum, the wind speed at 81 m decreased to 1 m s$^{-1}$
and for the same time the virtual potential temperature profile showed a mild gradient of about 1.2 K. This mild temperature increase with height above 41 m indicates that the decoupling of the air above the forest is less strong, explaining why the $CH_4$ is measured subsequently at lower heights. Note that vertical transport for these situations is triggered by mechanical turbulence (generated by wind shear instabilities) or intermittent turbulence that, in the absence of a very strong thermal inversion, can transport $CH_4$ to the lower inlets and inside the canopy (Oliveira et al., 2018). During this night, the gradient was less
pronounced and it lasted for less time than in the night shown in Fig. 6a. At 07:30, $CH_4$ mixing ratios are very similar and the wind speed increase to more than 2.5 m s$^{-1}$.

In Fig 6c a night in which no gradient was measured is shown. For this night the coupling of the air above the canopy is shown by the almost constant $\Theta_v$ profiles for the three selected times of the night. The thermal gradient for the layer above the canopy is about 1.5 K. The $CH_4$ mixing ratios are very similar at all inlet heights for the full 24-hour period. During this night
the wind speed has both the largest values and the lowest variability during the course of the night. From these three cases, one can see that positive gradient events are associated with atmospheric stability and wind speed. However, as we show in the next section, the wind speed is not necessarily always weak for positive gradients. Therefore, a deeper analysis of the main atmospheric conditions under which positive gradients are found is presented next.

### 3.2 Atmospheric characteristics of positive $CH_4$ gradients

As discussed in the previous section, a deeper understanding of the atmospheric conditions during positive gradient events is needed. In Sect. 3.2.1 and 3.2.2 the analysis is performed using 30-minute data for both $CH_4$ mixing ratios and micro-meteorological variables. In Sect. 3.2.3 we analyze the $CH_4$ positive gradients for the stable boundary layer taking into account the turbulence regimes defined by Sun et al. (2012). The stable boundary layer hereafter be referred to as the nocturnal boundary layer (NBL). These turbulence regimes were identified above another Amazon forest canopy by Dias-Junior et al. (2017)
using 5-minute averages for wind speed and vertical velocity ($w$). Therefore in this analysis we use high-frequency data (10 Hz) for micro-meteorological variables but we averaged to 1-minute, in order to be consistent with recent practice in nocturnal boundary layer studies (Marht et al., 2013; Acevedo et al., 2016, 2019) and to more strictly filter out low-frequency submeso fluctuations (i.e. non-turbulent motions at scales smaller than those at the mesoscale) (Mahrt, 2009) in all wind components. $CH_4$ mixing ratios consist of 15-minute intervals because our sampling system is not designed for high-frequency





measurements, this is the highest temporal resolution we can obtain. The way in which we associated the turbulent regimes with $CH_4$ positive gradients is explained below.

### 3.2.1 $CH_4$ positive gradients and wind direction for 2013-2018

Associating $CH_4$ positive gradients with the prevailing wind direction provides information about potential source areas. We
have done this by using the Openair package in R developed by Carslaw and Ropkins (2012). This R package provides useful predetermined functions to interpret air pollution characteristics based on wind speed, wind direction and other variables. The analysis was performed at hourly and monthly time scales by calculating a mean $CH_{4_{grad}}$ for each bin conformed by wind direction and time of day (hourly) or wind direction and month (monthly), as shown in Fig 7. For hourly time scales, we find that for nighttime hours and when the mean $CH_{4_{grad}}$ is above 8 ppb the wind direction is within the range of 90 and 180
degrees, hereafter southeasterly (Fig. 7, left panel). Note that the wind directions ranging between 180-270 degrees, hereafter southwesterly, also show positive mean $CH_{4_{grad}}$ during nighttime hours but these are lower than those seen when the wind comes from the southeast. For lower or negative mean $CH_4$ gradients and other times of the day, mainly daytime, the wind direction does not show a dominant pattern.

At monthly time scales we observed similar behavior, with a dominant southeasterly wind direction for mean $CH_{4_{grad}}$ above
8 ppb and the dry season months of August and September. At this monthly time scale, during August and September, the southwesterly direction shows mean $CH_{4_{grad}}$ above 4 ppb, suggesting that this wind direction plays an important role in the positive $CH_{4_{grad}}$ events. Yet, it is very clear that large positive $CH_{4_{grad}}$ are mainly driven by southeasterly winds. Given that there is a prevailing pattern at monthly and diurnal time scales we can suggest that the potential source of the nighttime positive $CH_4$ gradient is associated with this wind direction. During dry season months the wind is more frequently coming from the
east whereas during the wet season there is a shift to northeasterly winds. *In-situ* measurements of wind direction confirm this pattern. The monthly mean wind roses (see Fig. A1) suggest that the wind is more frequently coming from the east and it is more likely to bring air masses from the southeastern areas of ATTO during June, July, August and September. These months have wind direction frequencies close to 20% with wind speeds ranging from 1 to 7 $m\ s^{-1}$. The mean wind rose plots for each hour of the diurnal cycle (see Fig. A2) indicate that after 15:00 local time, wind directions in the 90-180 degrees quadrant
become more important than in previous hours of the day, with frequencies of about 15%. After 18:00, the frequency increases at this wind direction until 23:00. These wind direction patterns together with the prevailing wind direction for the positive $CH_{4_{grad}}$ suggest that nighttime positive gradient are more frequent in the dry season due to the prevailing wind direction, being more likely to bring air masses from the potential source region located to the southeast of ATTO.

More information about the location of the potential source is given by the dominant wind speed during the nighttime
positive $CH_{4_{grad}}$. Keeping the conservative threshold value of 8 ppb used previously, which corresponds to the 88th percentile of the gradient ($CH_{4_{79m}}$ - $CH_{4_{4m}}$) distribution, we plotted the probability of measuring gradients above this percentile together with wind direction and wind speed at 81 m (see Fig. 8). From this we obtain some interesting information. First, the highest probability (50%) of having a gradient above 8 ppb is associated with a particular wind speed range, mainly from 4 to 5 m $s^{-1}$. This wind speed range is seen for different wind directions, with the two southern quadrants (from 90 to 270 degrees)



having a very similar probability of 40 to 50%, with a slightly higher conditional probability in the southeastern quadrant than the southwestern quadrant. For the northwest quadrant (270 to 360 degrees) the probability observed is lower than 40% but interestingly the wind speed range holds. Therefore, the gradients above 8 ppb, which are mainly occurring during nighttime, are associated not only with a range of wind directions but also with a particular wind speed range. We showed previously

that southeasterly winds are clearly bringing $CH_4$-enriched air masses, but there is also a 40% probability of having these enhancements (above 8 ppb) when the wind direction shifts to the southwesterly direction and also when it comes from the northwest with approximately 25% probability at wind speeds of 5 m s$^{-1}$. It is relevant to note that 68% of the positive $CH_{4_{grad}}$ occur at wind speeds within the range of 2 to 5 m s$^{-1}$ (not shown), which in the southern quadrants still have a probability of occurrence higher than 20% and lower than 50%.

Considering these results we can infer the following about the source of the nighttime positive $CH_4$ gradients. First, dominant large scale circulation patterns explain why large positive $CH_4$ gradients are more common during the dry season months, and this is because the wind arriving at ATTO is more likely to come from the source areas to the south of the site. Moreover, this also explains why large positive $CH_4$ gradients are more strongly associated with the southeasterly direction; wind direction frequency is larger for these months than for other months of the year. Second, large positive $CH_4$ gradients can also come from

other wind directions, but they are less frequent because the wind is less likely coming from those directions. Therefore, we identify a potential source of the nighttime signals with a probability of 40% to 50% to the south of the ATTO site (see Fig. 8). Here it is important to recall that these percentages are based on the conservative threshold of 8 ppb, analyses not shown here using a lower threshold yield a slightly higher probability. Third, nighttime average wind direction is more frequently coming from the southeast, explaining why positive $CH_{4_{grad}}$ are more strongly associated with these wind directions at this timescale.

### 3.2.2   Net radiation, sensible heat flux, friction velocity and thermal stratification for $CH_4$ positive gradients 2013-2018

In this section we assess the role of atmospheric stability on the nighttime positive $CH_4$ gradients. We used micro-meteorological data at 30-minute resolution collocated with the $CH_4$ mixing ratio data at ATTO, covering the time period between June 2013 and November 2018. The heights of the highest sonic anemometer and the highest air inlet for $CH_4$ mixing ratio measurements

differ by two meters, with the former at 81 m and the latter at 79 m. We assume that the effect of the two meters can be neglected and thus interpret all the 81 m data as valid for 79 m.

    To better understand the atmospheric characteristics under which nighttime $CH_4$ positive gradients occur, we use net radiation, sensible heat flux ($H$), friction velocity ($u_*$) and the virtual potential temperature ($\theta_v$) together with the data points with a $CH_{4_{grad}}$ above and below 8 ppb (see Fig. 9). The variability of these quantities is plotted for each hour of the night,

beginning at 20:00 until 06:00 (local time). In order to maintain consistency, we used the same 8 ppb threshold as in Sect. 3.1, but this time we clustered the other two classes ($CH_{4_{grad}} < $ -8 ppb and -8 $< CH_{4_{grad}} < $ 8 ppb) into only one: below-8-ppb. The latter provides more clarity in the interpretation, as we are strictly interested in the above-8-ppb class. Net radiation for the above-8-ppb class is more negative, indicating a stronger radiative cooling at ATTO (see Fig. 9a) when these episodes occur. Net radiation is less variable for positive gradients with lower mean and median values for all night hours. This association





can be explained because positive gradients are more frequent during the dry season and in particular in August, when there is less cloud cover, as can be inferred by our precipitation record (Fig. 2), and as reported in Andreae et al. (2015). Clear skies contribute with a more effective radiative cooling at the canopy top, leading to a stronger thermal inversion in the NBL. Net radiation estimates at ATTO provide indirect evidence that suggest a shallower NBL height during the dry season. Net radiation

values above the canopy are more negative during the dry season than during the wet season (see Fig. B1), suggesting a strong thermal inversion driven by large nighttime radiative cooling at the top of the canopy. For very stable nights at the ATTO site, Oliveira et al. (2018) found that at 81 m the turbulent fluxes were clearly less variable than at lower heights, indicating a very shallow NBL height.

Positive $CH_4$ gradients are associated with a less variable $H$ (see Fig. 9b), tending to values close to zero. For the above-8-

ppb class, the mean and median values of $H$ at the beginning of the night hours, mainly from 20:00 to 23:00, are slightly more negative than for the other class. After midnight these values tend to be closer to zero for the above-8-ppb class, suggesting that the nocturnal boundary layer is more stable for positive gradient events. The variability of both classes between 20:00 and 23:00 is not as different as for the rest of the hours, but still is slightly lower for the above-8-ppb class. Considering the height at which the $H$ is zero, a proxy to infer the NBL height, one could say that positive $CH_4$ gradient events are associated with a very

shallow nocturnal boundary layer, close to 81 m. The seasonal difference in NBL height was studied by Carneiro (2018) during the GoAmazon campaign (Martin et al., 2016), using a Ceilometer, a Lidar and a SODAR, among other instruments. Carneiro (2018) found that the time to erode (total erosion is considered to be when sensible heat flux and net radiation become positive) the nocturnal boundary layer is larger during the wet season, suggesting a deeper NBL height as one of the reasons to explain this. It is worth noting that the Ceilometer sensitivity differs between instruments and it can also be affected by changes in

the background radiation (Wiegner and Geiß, 2012). The ceilometer used by Carneiro (2018) is a CL31 (Vaisala Inc., Finland) and the one at ATTO is a Jenoptik CHM15kx, used recently by Dias-Júnior et al. (2019) to determine the mixing layer depth using only daytime backscatter profiles. In another study with the Jenoptik CHM15kx, Wiegner and Geiß (2012) found that the lowest detectable mixing height is around 150 m, therefore nights with a NBL height lower than this might not be well captured. Furthermore, the study was conducted over pasture which has different roughness and radiative characteristics compared to

old-growth forest, and as such the results cannot be extrapolated to ATTO. Nonetheless the study provides valuable information about seasonal differences of the NBL height. We assume that the estimates given by Carneiro (2018) show a realistic range of variability, as the study makes a thorough comparison between several instruments.

Additionally, positive gradients are associated with low friction velocity ($u_*$) variability as well as low mean and median values for all nighttime hours (see Fig. 9c). The nighttime trend for $u_*$ starts with similar mean and median quantities for

both classes, above and below 8 ppb. At 21:00 and afterwards, positive gradients above 8 ppb show lower values and a more compact distribution. As a measure of mechanical turbulence, low $u_*$ values suggest that positive gradients occur mainly at low turbulence intensity. This finding is not surprising as strong stability and the absence of turbulence can lead to accumulation of trace gases in the NBL (Stull, 1988; Fitzjarrald and Moore, 1990) due to reduced vertical mixing. However, under this common assumption and considering that the NBL above the canopy can attain shallow depths, one would expect to measure

the accumulation of trace gases at least at the other inlet heights above the canopy or at inlet heights closer to the canopy, where





the possible source could be located, but for many of the positive gradients events this is not the case. The $CH_4$ signal that arrives at the uppermost inlet (79 m), driving a positive gradient, is often not seen at lower inlets. This is very often the case for the inlet at 38 m but less so for that at 53 m, indicating that the $CH_4$ rich air is advected within a layer that includes the 79 m inlet and sometimes the 53 m inlet, but not those below 53 m. Having low friction velocity values and considering that the

dominant wind speeds at which positive gradients have more probability of occurrence are between 2 to 5 m s$^{-1}$ (see Fig. 8), suggest that $CH_4$ signals are transported mainly by horizontal non-turbulent motions, which are probably formed by inactive turbulence mainly seen at the upper layers. Such inactive turbulence contributes very little to the generation of turbulence as was indicated by Högström (1990). Acevedo et al. (2016) showed that there is a wind speed threshold at a certain height, referred to as *crossover threshold*, at which the NBL switches from a vertically decoupled to a coupled regime. Using data

from a 30 m tower, they found that for a wind speed close to 2 m s$^{-1}$ (at 1 m), the 30-m layer was fully vertically coupled. Under decoupled conditions $u_*$ increases with height and at each individual level this quantity is constant before the *crossover threshold*. Under a coupled NBL after the *crossover threshold*, $u_*$ converges to similar values at all levels and increases with wind speed. Therefore, we believe that under positive gradients the NBL is strongly decoupled, with a layered profile of $u_*$ until 80 m, but not necessarily with low wind speeds at this height.

The apparent decoupling of the above-canopy air for positive gradients suggests a very strong thermal stratification below 79 m. The difference of $\theta_v$ between 81 and 36 m is slightly higher for the above-8-ppb class than for the below-8-ppb class (see Fig. 9d). The difference between the median values of the two classes is approximately 0.5 K for all night hours, hinting at a relatively faster radiative cooling at the top of the canopy for the above-8-ppb class. The variability of this difference is very similar between both classes, although the above-8-ppb class is more skewed towards positive values. The difference between

mean and median values for both classes is small, but the fact that we see a systematic difference at all night hours, strengthens the argument that thermal stratification is stronger when positive gradients occur than for the other class.

Recalling the aim of this section, i.e. to show general atmospheric characteristics for the positive gradient events, here we summarize the general characteristics of positive gradients:

1. The source region is south of ATTO. In particular the southeastern region has the highest probability (between 40-50%)
of bringing air masses that result in positive gradients.

2. Positive gradients are associated with less variable $H$ and possibly a lower NBL height.

3. Low $u_*$ values indicate reduced mechanical turbulent motion and strong stability for positive gradient events.

4. Thermal stratification is stronger for positive gradient events, supporting the idea of a decoupled regime during these events.

**3.2.3   Association of positive $CH_4$ gradients and NBL turbulence regimes**

In this section we proceed to assess the role of stability regimes in the positive gradient events. For this purpose, we focus on 6 months of 2014 (March, April, May, July, August and September). We have selected these months to focus purely on wet and





dry seasons, and discarded June because it is considered a transition month based on precipitation. For these months we used high-frequency (10 Hz) data at ATTO to define the turbulence regimes of the NBL based on Sun et al. (2012). These regimes are then classified into two classes, the same as we have described before: above 8 ppb and below 8 ppb. This classification is performed using the highest temporal resolution of $CH_4$ mixing ratio data, which is 15 minutes. We assumed that the gradient

is constant for each 15-minute interval. For example, if the $CH_4$ gradient is 5 ppb at 21:00, we use the same value for every minute until 21:14. We have shown before in Sect. 2.2 that our $CH_4$ mixing ratio measurement system provides independent samples for each inlet height about every 15 min.

As shown in Sect. 3.2.2, positive gradients occur under a decoupled regime, namely in the absence of full vertical connection within the tower layer above the canopy (Acevedo et al., 2016). Given these facts, the questions that arise with the information

we have about the occurrence of positive gradients are 1. Are positive gradient events associated with a particular regime of the NBL? If so, 2. What can we infer about nocturnal transport mechanisms by which enhanced $CH_4$ signals are brought to the 79 m inlet using *in-situ* high-frequency micro-meteorological data?

To answer the first question, we have defined the NBL turbulence regimes following the work of Sun et al. (2012), which was further applied by Dias-Junior et al. (2017) at another site within the Amazon forest. In Fig. 10a the standard deviation of

the vertical velocity ($\sigma_w$) is plotted as a function of mean horizontal wind speed ($U$) for all the data available in 2014 without differentiating between gradient classes. Here, one can see that the regime transition (between regime 1 and 2) occurs at the wind speed bin of 5.8 m s$^{-1}$, which comprises the range between 5.6 and 6.0 m s$^{-1}$. At this threshold the uppermost quartile of $\sigma_w$ exceeds 0.5 m s$^{-1}$ and the slope defined by median values changes notably. Therefore, we define this bin as the wind speed threshold that marks the transition from regime 1 to regime 2. In regime 1, turbulence is produced by local shear at low

wind speed and low $\sigma_w$. In regime 2, $\sigma_w$ increases with wind speed and bulk shear in the NBL triggers turbulence (Sun et al., 2012). Subdividing this further into the gradient classes above and below 8 ppb (Fig. 10b), we observe that the variability of $\sigma_w$ close to the wind speed threshold (bins 5.8 and 6.5 m s$^{-1}$) is reduced for the above-8-ppb class. The upper quartile of $\sigma_w$ for the above-8-ppb class exceeds 0.5 m s$^{-1}$ only up to 6.5 m s$^{-1}$. The median $\sigma_w$ for both classes display a similar range of variability at 7.5 m s$^{-1}$. The above-8-ppb class has a different distribution in the same wind speed bins close to the wind speed

threshold, having lower median values of $\sigma_w$. This finding implies that vertical motions are even more suppressed at 81 m for the above-8-ppb class, regardless of the wind speed exceeding the threshold value.

Given these facts, we can answer the first question above, and associate positive gradients with the more stable regime 1: low $\sigma_w$ even at wind speeds exceeding the threshold value, identified without separating into gradient classes. In other words, the wind speed threshold under positive gradient events is shifted to higher wind speed, maintaining low $\sigma_w$ values. To account

for wind speed variations that could affect our assumption of constant $CH_4$ mixing ratios for every 15-minute window, we have filtered out the 15-minute time periods in which the difference between the maximum and minimum wind speed is larger than 0.5 m s$^{-1}$ (Fig. 10c). After filtering, the dependence of $\sigma_w$ on wind speed shows a similar pattern for each of the classes. In general, $\sigma_w$ is less variable for positive gradients events. However, the difference in wind speed threshold is even more evident at the wind speed bins of 5.9 and 6.1 m s$^{-1}$. Conclusively, positive gradients are associated with the stronger stability of regime

1. According to Sun et al. (2012), turbulence in regime 1 is weak and is controlled by vertical temperature gradients, in line





with our finding of a stronger thermal stratification for positive gradients shown before. In regime 1 eddies generated at the observation height triggered by local shear do not come into contact with the ground (Sun et al., 2012). This is because, as stated by Sun et al. (2012), the length scale of the local shear is smaller than the observation height.

More evidence about the dominant regime of the NBL when the positive gradients occur is given in Table 2. The statistics are based on only nighttime data, namely a time period between 20:00 and 06:00 (local time). For the 1-minute data we used a record of 6 months in 2014, the same months as above. The values shown are affected by very large outliers, therefore for defining the minimum and maximum limits of the 1/L classes in Table 2, we rely on the upper and lower quartiles $\pm$ 1.5 * inter-quartile range (IQR). In this way we are leaving some data points unclassified, which is why the percentages in Table 2 do not add up to 100% but still cover 77.1% of the positive gradient events during this period of time. Almost half of the positive gradient events, 49.1%, occur during moderately stable or very stable conditions, 6.7% under neutral conditions and 21.4% under moderately unstable and very unstable conditions. Having 49.1% of the nighttime episodes under moderately stable or very stable conditions is in line with the association with regime 1 in the turbulence regime analysis. The fact that 21.4% of the events occur under moderately unstable or very unstable conditions indicates that a considerable portion of the positive gradient events can also occur even when vertical motions are present. These situations are very likely taking place when the wind direction coincides with that of the source location.

As shown before, half of the positive gradients occur under a moderately stable or very stable NBL, with low values of $\sigma_w$ (even at relatively large horizontal wind speeds) and low $u_*$. As a result, one can infer the absence of vertical motion driven by nighttime turbulent mixing at the tower location. We can only attribute this lack of vertical mixing above the canopy to the tower location. Therefore, we have to separate the NBL conditions at the tower and at the potential source location. Unfortunately, we only have measurements at the tower and it is not realistic to measure at all possible source locations. Thus, and elaborating further on the second question established above, the transport mechanisms can be divided into 1. those responsible for vertical transport of $CH_4$ at the source location and 2. those responsible for the horizontal advection bringing the $CH_4$ signals to the tower. Nocturnal vertical exchange, the mechanisms referred to in 1., can be driven by intermittent turbulence (Acevedo et al., 2006; Oliveira et al., 2018), gravity waves (Fitzjarrald and Moore, 1990), katabatic or drainage flows (Goulden et al., 2006; Tóta et al., 2008; Araújo et al., 2008; Tóta et al., 2012) and nocturnal land-river breezes (De Oliveira and Fitzjarrald, 1994; Sun et al., 1998). For the horizontal transport of $CH_4$ towards the tower, we have discarded a nocturnal low-level jet since these events are commonly associated with high wind speed, high friction velocities and weak to unstable conditions (Karipot et al., 2008), whereas positive gradients coincide with low $u_*$ and a broad range of wind speeds between 2 and 5 m s$^{-1}$, and they are not specifically associated with large wind speed. Therefore, we believe that horizontal transport of $CH_4$ from the source location to the tower is driven by horizontal advection of the prevailing wind, bringing methane-rich air when the wind direction coincides with those shown in Fig. 7. More about these mechanisms will be discussed in the next section.





### 3.3 Potential sources and transport mechanisms of positive nighttime $CH_4$ gradients

#### 3.3.1 Constraining the potential sources

Based on the predominant wind direction of the positive gradients, we propose a potential source of the nighttime $CH_4$ signals at the 79 m level. First in importance and most dominant is the southeastern quadrant, second in line is the southwestern
quadrant (see Fig. 7 and 8). In these directions lies the Uatumã River, which we believe is the main and most likely source (but not unique) of the positive gradients seen at ATTO. In addition to the natural $CH_4$ production by this river, we propose two additional sources that could add up to the natural production and enhance $CH_4$ degassing from this water body. The first one is related to an enhanced $CH_4$ concentration downstream of the Balbina reservoir. As it was shown by Kemenes et al. (2007), the Balbina reservoir not only leaks methane at the turbine intake due to depressurization but it also enhances $CH_4$
concentrations downstream along the Uatumã. Kemenes et al. (2007) found that $CH_4$ concentrations in the Uatumã River decreased gradually until 30 km below the Balbina reservoir and remained constant for the next 70 km. It is worth noting that the $CH_4$ concentrations in the river, reported by Kemenes et al. (2007), were on average 24 $\mu$M, three orders of magnitude higher than the average background value (0.05 $\mu$M) for Amazonian rivers (Richey et al., 1988; Kemenes et al., 2007). The second additional source is very likely decomposition of dead flooded forest stands downstream the Balbina reservoir along
the Uatumã River. The dead stands are a consequence of the Balbina damming, which has changed the natural flooding pulse along the river, causing massive mortality of flooded forest trees along a 80 km stretch downstream of the dam (Resende et al., 2019). These dead stands were mapped recently by Resende et al. (2019), and their spatial distribution coincides with the wind directions associated with the positive gradient events (see Fig. C1).

    Here we suggest the Uatumã River as the dominant source of nighttime $CH_4$ positive gradients, but we want to make clear
that $CH_4$ could also be produced in the topographic depressions or valleys that form the drainage network surrounding ATTO. We have mentioned earlier that unpublished measurements of atmospheric $CH_4$ mixing ratios performed recently in one of these valleys indicate a nighttime increase of $CH_4$ within the canopy. Based on Junk et al. (2011), these depressions can be classified as *wetlands subjected to unpredictable, polymodal flood pulses fed by rainwater*, with low nutrient availability compared to the plateaus. The flooding dynamics are driven by flash flood pulses after precipitation events, which can flush out
a large amount of the organic material available for decomposition (Wittmann F., 2019, personal communication). Therefore, these depressions were assumed to have low potential of producing $CH_4$. Nevertheless, the accumulation seen for $CH_4$ mixing ratio measurements performed during May and June in 2019 indicate that either $CH_4$ is transported to the lowest part of the valley and accumulates during the night, or that $CH_4$ is produced at the valley and accumulated *in-situ*. Therefore, we suggest two potential $CH_4$ sources. First, the Uatumã River and the downstream dead stands as the most likely one and second, the
valleys in the ATTO surroundings.

    The question that arises here is, why do we see a seasonal pattern in the positive gradients, being more frequent and predominant during the dry season? Considering the Uatumã River as the main source, we can explain this seasonality based on the effectiveness with which methane is degassed from the river to the atmosphere and the prevailing atmospheric conditions that drive atmospheric transport from the river to ATTO. During the dry season, when the river levels are low, degassing is





more effective due to lower hydrostatic pressure, rendering the ebullitive pathway more efficient due to a shorter water column, which also reduces the probability of oxidation (Sawakuchi et al., 2014). Therefore, during the dry season the $CH_4$ produced in the river sediment plus that added by the Balbina reservoir could be more effectively emitted to the atmosphere. It is important to note here that the suggested source coming from the anaerobic decomposition of the dead stands of flooded forests or *igapós*,

could also be affected by the shorter water column during the dry season. However, in terms of enabling anaerobic conditions in the sediments, we have to make a distinction between floodplain and riverine environments, as these could have different responses to flooding. Anaerobic conditions in floodplains soils seem to follow the established idea that with a higher water level there should be higher methane emissions (Kaplan, 2002; Bloom et al., 2012; Melton et al., 2013; Ringeval et al., 2014; Bloom et al., 2017), which are either diffused or transported by ebullition or trees (Pangala et al., 2017) to the atmosphere. In

contrast, sediment in rivers could always be anaerobic with the potential to produce methane regardless of the season. What is really affected by a deeper water column in a river is the time for $CH_4$ to be oxidized and the increased hydrostatic pressure that can inhibit the ebullitive pathway (Sawakuchi et al., 2014). Given the aforementioned, we can now link these potential sources along the Uatumã river and its seasonal pattern with the dominant atmospheric conditions during the dry season months. During these months and in particular in August, as we have shown in the last section, the frequency with which the wind brings air

from the southeast to ATTO is higher than during the wet season months, when the prevailing wind direction is due northeast (see Fig. A1). Therefore, the probability of advecting methane-rich air emitted in the Uatumã River area and potentially the valleys along that same direction is also higher during the dry season.

The Amazon River was discarded as a potential source even though it coincides with the wind direction found for the positive gradients. The Amazon River is 120 km southeast of ATTO, which means that a strong $CH_4$ emission into the nocturnal

boundary layer will have to be advected at a wind speed of 6 m s$^{-1}$ to reach ATTO in five hours. This might be possible on some occasions, but as we saw before (Fig. 8) positive gradients are associated with wind speeds between 2 and 5 m s$^{-1}$. Moreover, 80% of the wind speed for nighttime positive gradients is below 4 m s$^{-1}$. In addition, we calculated the distance to the $CH_4$ source by time-integrating the wind speed at 81 m from 20:00 (beginning of the night) until the first occurrence of a positive gradient (> 8 ppb). The distribution of these distances is shown in Fig. 11. The distances with more counts are below

50 km, 90% of the data points fall below 100 km, and 80% of the data points are below 72 km. Given these facts, the Amazon River is not considered as a potential source for the nighttime $CH_4$ positive gradients.

### 3.3.2 Atmospheric transport mechanisms from the source

We have linked the seasonal pattern seen for the positive gradients to the location of the $CH_4$ sources and the driving atmospheric conditions responsible for horizontal transport. Now we proceed to describe in more detail the mechanisms by which

$CH_4$ is transported vertically at the source location. As we lack *in-situ* measurements at the proposed source locations, we build upon previous studies that have discussed mechanisms of vertical exchange of scalars in the NBL and assume that methane-rich air is transported vertically, from the surface to the upper layers of the NBL, by these mechanisms at the source location. Earlier in Sect 3.2.3, we listed some of these mechanisms, here we will address them while taking into account their probability of occurrence at the Uatumã River and at the valleys surrounding the tower. Intermittent turbulence mainly occurs as top-down





bursts (Sun et al., 2012) that can connect the upper layers of the NBL with the canopy and even penetrate the upper part of it, as was shown to be important for ozone and $CO_2$ fluxes at the ATTO site by Oliveira et al. (2018). However, as intermittent turbulence (or regime 3 events as defined by Sun et al. (2012)) are mainly observed as top-down intrusions, we discard it as a vertical transport mechanism at the source locations, given that the source of $CH_4$ has to be at the surface. Gravity waves

were shown to be responsible for transporting mass out of the sub-canopy layer during strong stability conditions and clear nights (Fitzjarrald and Moore, 1990), yet Cava et al. (2004) suggested that wave motions do not play an important role in scalar transport based on the fact that one wave period had zero mean flux. Therefore, it is difficult to firmly associate gravity waves with a vertical transport of $CH_4$ at the valleys or at the Uatumã River. Having discarded these mechanisms, we believe that land breezes and drainage flows are the most probable mechanisms inducing vertical transport at the Uatumã River and at the

valleys close to ATTO. During the night, due to differential radiative cooling over land and water, the river is warmer than the forest, leading to slightly warmer air over the water. This leads to a breeze from the land to the water that can transport water vapor and trace gases from the forest to the river. Such breeze (from land to water) was observed over the Balbina reservoir by Vale et al. (2018), finding a nocturnal accumulation of $CO_2$ over the water. Measurements during the ABLE experiment showed that the Amazon River could be 6 degrees warmer than the forest (De Oliveira and Fitzjarrald, 1994). Considering that

black water rivers, such as the Uatumã are warmer than white water rivers, like the Amazon River, it is very likely that the Uatumã River is warmer than the surrounding forest. The temperature difference produces a pressure gradient as warm and less dense air moves vertically over the river. These air parcels can vertically transport $CH_4$ to upper layer of the NBL. This mechanism was described as a "chimney effect" in the study of Sun et al. (1998), in which they found that water vapor, ozone and $CO_2$ can be vertically transported by these events over a lake. The width of the lake in their study was about 10 km, and

the width of the largest open water flooded areas along the Uatumã River southeast of ATTO, the Lago Cumateúba and Lago Araçatuba, are 3.5 km and 1.8 km.

The vertical transport mechanism at the valleys could be slightly different from that over the river. We believe that updrafts driven by air convergence forced by drainage flows can lead to vertical transport. Drainage flows in the Amazon were inferred by Goulden et al. (2006) using *in-situ* measurements and remote sensing imagery and although vertical transport mechanisms

at the lower topographic areas were not addressed, they suggest that the air at the center could be well-mixed. Later, Araújo et al. (2008) showed that nocturnal katabatic flow from the plateau to the valley not only affects the horizontal distribution of $CO_2$ mixing ratios along a topographic gradient but also the vertical profile of $CO_2$ mixing ratios over the valley. They had tower measurements on the plateau, on the slope and at the valley floor. With this setup they could confirm the occurrence of a shallow convergence zone over the valley that breaks down the thermal inversion and transports air vertically from the valley

floor to the layers above. Araújo et al. (2008) suggest that the air might sink over the slope to maintain the local circulation, but the katabatic or drainage flow at ATTO could be more pronounced due to steeper slopes compared to those in the Araújo et al. (2008) study. On a different study in the Amazon forest (Tóta et al., 2012), drainage flows were found to occur above the canopy, moving from the plateau to the valley as in Araújo et al. (2008), but in the sub-canopy layer they observed upward (anabatic) flow during nighttime. If this process occurs at the ATTO site despite its steeper topographic gradients, it would not





explain the observed positive $CH_4$ gradients. Such a mechanism would instead result in negative $CH_4$ gradients as the $CH_4$ signal would arrive first to the 4 m inlet, which is not the case for positive gradients.

Due to the lack of measurements at the valley, we are limited to the studies we have cited previously and it is difficult to identify a dominant mechanism for vertical transport at these sites. However, we suggest the Uatumã River as the most likely
$CH_4$ source area, with possible contributions from the valleys surrounding ATTO. In the next section we discuss why biomass burning does not explain the positive gradients even though the timing suggests it.

### 3.3.3  Rejecting biomass burning influence on positive $CH_4$ gradients

The timing of the biomass burning season coincides with the dry season in the Amazon region (Gatti et al., 2014; van der Laan-Luijkx et al., 2015; Aragão et al., 2018), thus one could think that $CH_4$ from combustion is responsible for the positive
gradients presented here. During biomass burning, in particular in the case of incomplete combustion, $CH_4$ is co-emitted together with carbon monoxide (CO), amongst other gases and particles (Akagi et al., 2011; Kirschke et al., 2013; Andreae, 2019). Because CO is emitted in large quantities and due to its low background mixing ratio, this species is considered a good proxy for biomass burning and it can be used as a reference to get an idea of enhancement ratios due to fire emissions. Figure 12 presents a scatter plot of the 30-minute $CH_4$ and CO mixing ratios for nighttime measurements (at 79 m) during
which a positive gradient was observed and the wind was coming from the southern quadrants. Here, we observed large CO mixing ratios (>200 ppb), suggesting that for these measurements we could have sampled air with biomass burning signals. However, these data points represent only 10% of the data shown. Moreover, biomass burning typically produces $CH_4$ in a certain emission ratio to CO, dependent on the fuel type Andreae and Merlet (2001); Andreae (2019). These reference ratios are plotted as slopes in Fig. 12 and it can be seen that very few points that fall on the reference slopes and the $CH_4$ mixing
ratios are substantially enhanced compared to those of CO, indicating an additional source of $CH_4$ seen for all wind direction. The mean CO mixing ratio during nighttime in the dry season at 79 m ($140 \pm 1.6$ ppb) is on average 33 ppb higher than during the wet season ($107 \pm 1.25$ ppb), suggesting that during the dry season we observe a "background enhancement" of CO mixing ratios at ATTO. Therefore, we can conclude that $CH_4$ measurements at ATTO during the dry season will always have a contribution of biomass burning, but as Fig 12 shows this "background enhancement" of CO can not completely explain
the additional $CH_4$ of the positive gradients. Note that most of the data points are grouped below 200 ppb for CO, suggesting that positive gradients occur at CO mixing ratios close to the mean dry season mixing ratio. Given these facts, we believe that positive gradients have a minor contribution of $CH_4$ from biomass burning, but the magnitude of the $CH_4$ enhancement relative to the CO mixing ratios needs to have an additional source.

Therefore, the challenge constraining to what extent fire emissions affect our $CH_4$ relies on identifying a clear and distinctive
fire plume that provides information on not only the $CH_4$:CO but also the ratio relative to other species. Considering that black carbon (BC) particles are emitted in smoldering and flaming fires (Andreae and Merlet, 2001) together with CO and $CH_4$, we use BC measurements to gain more insights about the influence of fire emissions on the full $CH_4$ signal at ATTO. Following the same approach as in Fig 12, the BC:CO in Fig 13 suggests a more clear fire signal for the ESE and SSE wind directions. Note that the time period used in this plot differs form that in Fig 12 and contains fewer data points, yet some of those points fall on





the reference slopes of Andreae (2019). In general terms, even though fire signals are measured at ATTO during nighttime for positive $CH_4$ gradients as suggested by Fig 13, the nighttime $CH_4$ enhancements at 79 m are not fully explained by combustion. Most of the data points in the ESE wind direction match the reference slope for biofuel burning, which is defined by Akagi et al. (2011) as *biomass used as a domestic or industrial energy source*. Hence, biofuel burning is associated with human activity and according to the comprehensive study of Pöhlker et al. (2019), in the ESE direction there is substantial fire activity, the rainforest has suffered more fragmentation and degradation and there are more settlements. It is worth recalling that here we are focusing on nighttime data when stable atmospheric conditions prevail, therefore the BC associated with biofuel burning might come from nearby settlements. For other directions the fire signal is not so evident and for some data points the CO mixing ratio is very high. This can be seen for all wind directions, and we believe that this can be explained by a possible weakening of the BC:CO due to deposition of BC (Saturno et al., 2018). In summary, despite being able to detect a clear fire signature, $CH_4$ enhancements are too high relative to CO suggesting that an additional source is needed to explain the positive $CH_4$ gradients enhancement during nighttime.

## 4 Conclusions

In this paper we have studied $CH_4$ mixing ratios measured at ATTO. We showed that during the dry season, mixing ratios are on average higher at the top of the 80 m tower than at the lower inlet heights. We have defined these events as positive $CH_4$ gradients based on the condition: $CH_{4_{79m}}$ - $CH_{4_{4m}}$ > 8 ppb, which was applied to 6 years of continuous measurements at 30-minute resolution to classify our measurements. The $CH_4$ positive gradients of the dry season are associated with very strong $CH_4$ signals measured at 79 m, that occur more frequently during nighttime. Nighttime positive gradients ($CH_{4_{79m}}$ > $CH_{4_{4m}}$) are more frequent during July, August and September, occurring 43% of the time during nights in August, the month with the highest mean nighttime gradient (9.7 ± 21 ppb). At the diurnal time scale, we found that the amplitude of the mean diurnal cycle of the gradient during the dry season is four times larger than that of the wet season due to the strong nighttime positive gradients. The dominant wind direction for these nighttime episodes, at monthly and diurnal time scales, is southeast of ATTO. In addition, this direction has the highest probability, 50%, of bringing air that will cause a positive gradient (> 8 ppb). This probability was also found to be linked to a wind speed range between 4 and 5 m s$^{-1}$. This wind speed was associated with positive gradients for other wind directions as well, but with a lower probability. The dominant atmospheric conditions under which positive gradients occur are moderately and very stable conditions. When comparing the gradient data classes of > 8 ppb and < 8 ppb, we found that the first class can be described as having lower net radiation (it is more negative by -2 to -3 W m$^{-2}$, see Fig. 9), less variable sensible heat flux, low friction velocity (< 0.3 m s$^{-1}$) and a strong thermal inversion above the canopy. Further analysis of high-frequency micro-meteorological data suggests that positive gradients are associated with regime 1 of the nocturnal boundary layer, in which turbulence is weak, controlled by temperature gradients and generated by wind shear (Sun et al., 2012).

We discarded biomass burning as the main driver of the positive gradients, and have shown that the Uatumã River is very likely the most important source, due to the coincidence with the dominant wind direction. In addition, we suggest that two

additional $CH_4$ sources might enhance the natural emissions from the river area. The first one is a Balbina-reservoir-driven increase in $CH_4$ concentrations in the river (Kemenes et al., 2007), and the second possible source of $CH_4$ is due to anaerobic decomposition of dead stands of flooded forest along the Uatumã River downstream of the reservoir (Resende et al., 2019). The atmospheric transport mechanisms were divided into those responsible for horizontal advection of $CH_4$ from the source

locations to the ATTO site, and those that transport air vertically from the source location to the upper layers of the nocturnal boundary layer. We suggest that vertical transport over the Uatumã River results from differential radiative cooling of the forest and the water, producing a horizontal pressure difference that causes an upward displacement of air parcels over the river and transporting $CH_4$ aloft. These air parcels are then advected by the prevailing horizontal wind towards the ATTO site and subsequently measured at the 79 m level.

In the near future the 325 m tower will be fully equipped, providing valuable information in terms of $CH_4$ mixing ratios and meteorological variables which will enable us to study if the positive gradient extends to upper layers of the nocturnal boundary layer. We will be able to assess the influence of the residual layer and the height of the nocturnal boundary layer in our $CH_4$ measurements. To better understand local circulation and its effect on vertical $CH_4$ transport, we strongly recommend performing profile measurements at the river and in nearby valleys with emerging measurement techniques, such as unmanned

aerial vehicles. High resolution atmospheric transport models, such as the Weather Research Forecast for greenhouse gases (WRF-GHG), could also help to understand and either reject or confirm the mechanisms mentioned here. Furthermore, an upcoming campaign at ATTO specifically aims to determine the isotopic signature of the $CH_4$ mixing ratio during a positive gradient event, providing more accurate information about the $CH_4$ source.

*Data availability.*   All data used in these study are stored in the ATTO databases at the Max Planck Institute for Biogeochemistry and the
Instituto Nacional de Pesquisas da Amazônia. The $CH_4$ atmospheric mixing ratios are available on request with Jošt Lavrič (jlavric@bgc-jena.mpg.de). The micrometeorology dataset is available on request with Alessandro C. de Araújo (alessandro.araujo@gmail.com) and the black carbon data with Christopher Pöhlker (c.pohlker@mpic.de). The data generated at ATTO is becoming progressively available from http://attodata.org/.

*Author contributions.*   SBB and CG designed the study and wrote the manuscript with assistance of JM, GF and HvA. JL and DW maintain
the greenhouse gas measurement system at ATTO and provided the $CH_4$ data. ACA, MOS, PRT operate and maintain the micrometeorology equipment at ATTO and provided the data which was fundamental for this study. AFR provided the flooded forest dead stands data. CQDJ, PSO, MS and OCA contributed with the analysis and discussion of Sections 3.2 and 3.3. All co-authors contributed to the final manuscript.

*Competing interests.*   The authors declare that they have no conflict of interest.





Special issue statement. This article is part of the special issue "Amazon Tall Tower Observatory (ATTO) Special Issue".

*Acknowledgements.* We thank the Instituto Nacional de Pesquisas da Amazonia (INPA) and the Max Planck Society for continuous support. We acknowledge the support by the German Federal Ministry of Education and Research (BMBF contracts 01LB1001A and 01LK1602A) and the Brazilian Ministério da Ciência, Tecnologia e Inovação (MCTI/FINEP contract 01.11.01248.00) as well as the Amazon State University (UEA), FAPEAM, LBA/INPA and SDS/CEUC/RDS-Uatumã. We also want to acknowledge the International Max Planck Research School for Global Biogeochemical Cycles (IMPRS). We want to thank Paulo Artaxo for his valuable feedback regarding the biomass burning signals. We thank Florian Witmann for his comments about the hydrology regime in the ATTO surroundings. Furthermore, we acknowledge the people coordinating the scientific support at ATTO, in particular Susan Trumbore, Alberto Quesada and Bruno Takeshi. Finally, we want to thank all the personnel at the research site involved in technical and logistical support, specially Reiner Ditz, Andrew Crozier, Stefan Wolff, Leonardo Ramos de Oliveira, Nagib Alberto de Castro Souza, Roberta Pereira de Souza, Amauri Rodriguês Pereira, Hermes Braga Xavier, Wallace Rabelo Costa, Antonio Huxley Melo Nascimento, Uwe Schultz, Steffen Schmidt and Thomas Disper.





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



**Table 1.** Observational data used in each of the sections. We specify the time resolution and the period of time used. Met. data stands for meteorological data and refers to the variables described in Sect 2.3

| Sections | Data used | Time resolution | Period of time |
|---|---|---|---|
| 3.1, 3.2.1, 3.2.2 | CH$_4$ mixing ratios | 30 min | 2013-06 to 2018-11 |
| | Met. data | 30 min | 2013-06 to 2018-11 |
| 3.2.3 | CH$_4$ mixing ratios | 15 min | 2014: March, April, May, July, August, September |
| | Met. data | 10 Hz (avg. to 1 min) | 2014: March, April, May, July, August, September |
| 3.3 | CH$_4$ mixing ratios | 30 min | 2013-06 to 2018-11 |
| | CO mixing ratios in CH4:CO | 30 min | 2013-06 to 2018-11 |
| | CO mixing ratios in BC:CO | 30 min | 2014-01 to 2015-12 |
| | Black Carbon | 30 min | 2014-01 to 2015-12 |

**Table 2.** Percentage of nighttime positive gradient occurrences for different 1/L classes. The 1/L classes were defined based on Kruijt et al. (2000). Note that the percentages do not add to 100% because the upper and lower boundaries of the classes are based on Q1, Q3 $\pm$ 1.5 * inter-quartile range (IQR).

| | Positive Gradients Events (%) |
|---|---|
| Q1 - 1.5 * IQR < 1/L < -0.1 - Very unstable | 9.3 |
| -0.1 < 1/L < -0.01 - Moderately unstable | 12.1 |
| -0.01 < 1/L < 0.01 - Neutral | 6.7 |
| 0.01 < 1/L < 0.1 - Moderately stable | 26.7 |
| 0.1 < 1/L < Q3 + 1.5 * IQR - Very stable | 22.4 |





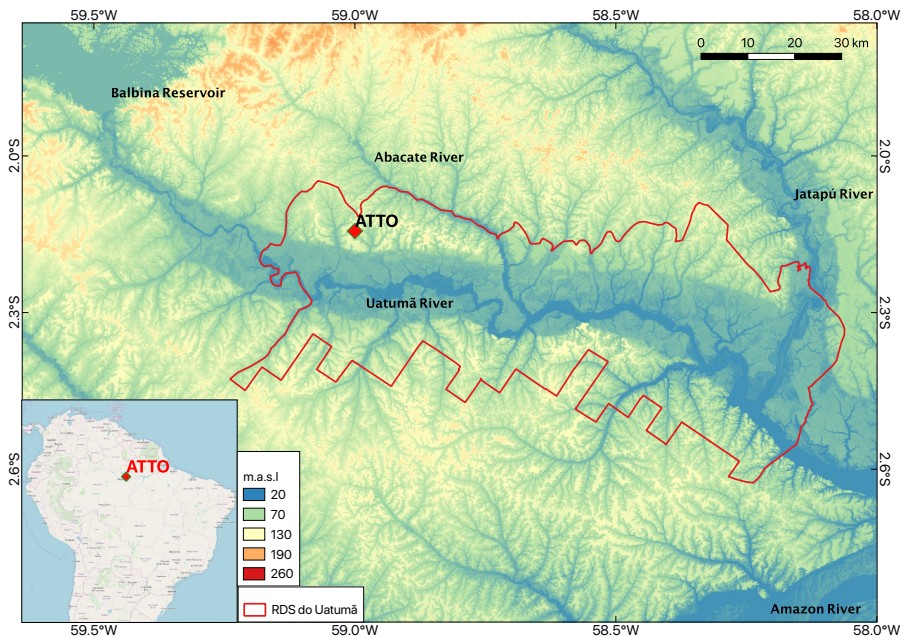

**Figure 1.** Location of ATTO relative to the continent. The topography, in the background, is based on the elevation model from the Shuttle Radar Topography Mission (NASA-JPL, 2013). The boundaries of the Uatumã Sustainable Development Reserve (USDR) are highlighted in the red polygon and the major rivers are labelled.

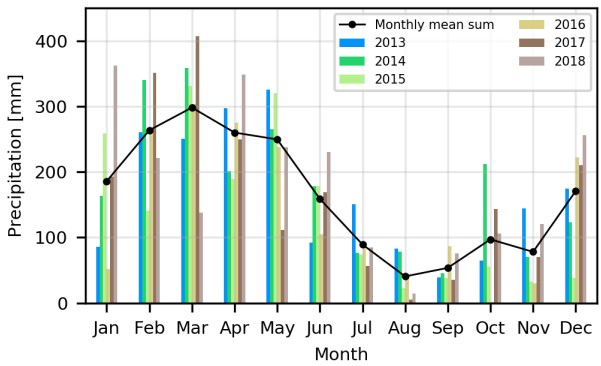

**Figure 2.** Monthly sums of precipitation for the period between 2013-2018 at ATTO.





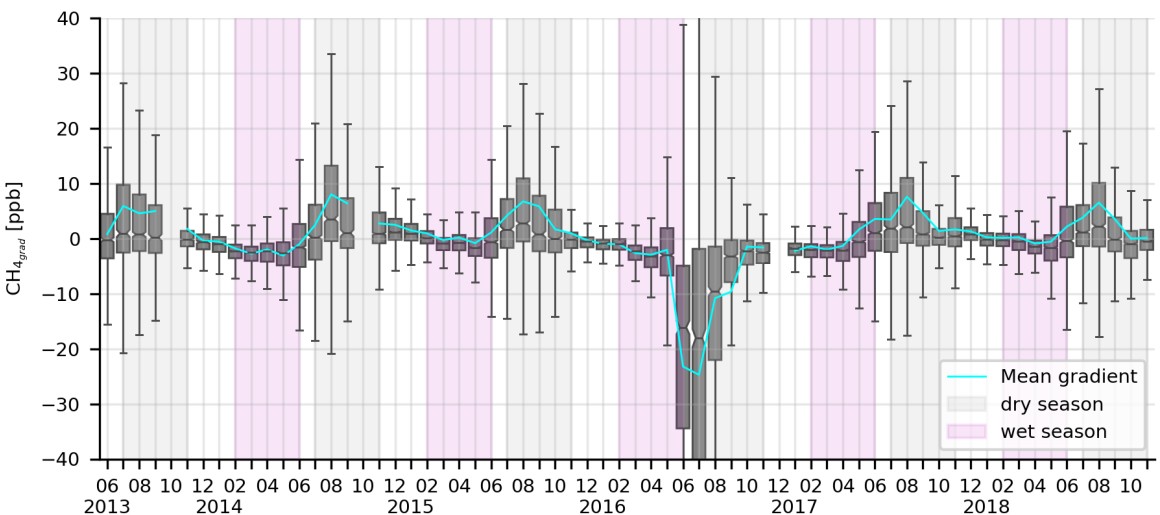

**Figure 3.** Monthly box-and-whisker plot of CH$_4$ gradient between the 79 m and 4 m levels. The box denotes the inter-quartile range (IQR), showing the median with a notched line. The whiskers range from Q1-1.5*IQR to Q3+1.5*IQR, with Q1 and Q3 being the 25th and 75th percentiles. The cyan line is the monthly mean gradient. The monthly statistics are calculated from half-hourly measurements at ATTO.





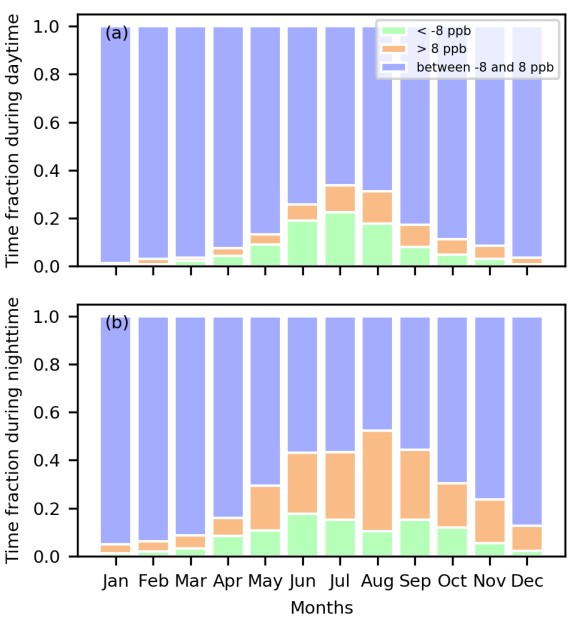

**Figure 4.** Time fractions for daytime (a) and nighttime (b) measurements in which $CH_{4_{grad}} > 8$ ppb, $-8 < CH_{4_{grad}} < 8$ ppb and $CH_{4_{grad}} < -8$ ppb. Nighttime is defined as between 20:00 and 06:00 and daytime as between 06:00 and 18:00, local time.

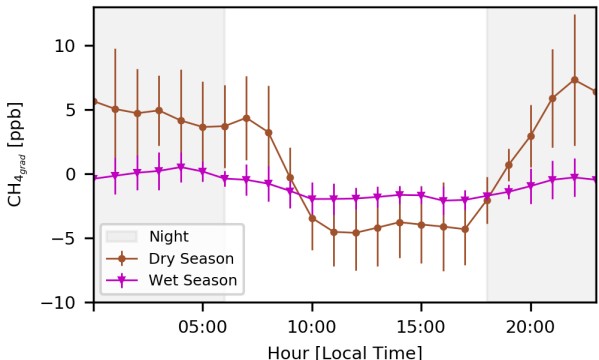

**Figure 5.** Mean diurnal cycle of the $CH_4$ gradient between the 79 m and 4 m levels. The period between June 2013 and November 2018 was used and separated by dry and wet seasons. The error bars show the standard deviations.




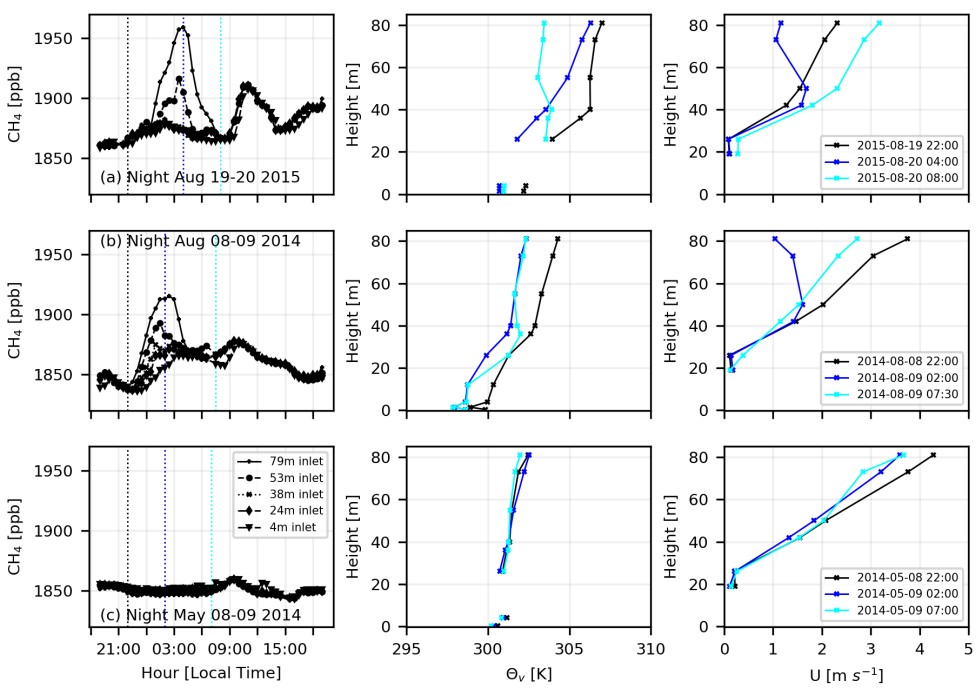

**Figure 6.** CH$_4$ time series for three selected nights (first column) showing a 24-hour interval, virtual potential temperature (second column) and mean wind profiles (third column) for three selected periods of the CH$_4$ time series. The times at which the profiles are plotted are highlighted on the CH$_4$ time series with the same colors. The canopy height is approximately 35 m. Note that due to instrument malfunctioning the $\Theta_v$ profiles are lacking data at 12 m. The wind speed profiles cover from 19 m to 81 m.



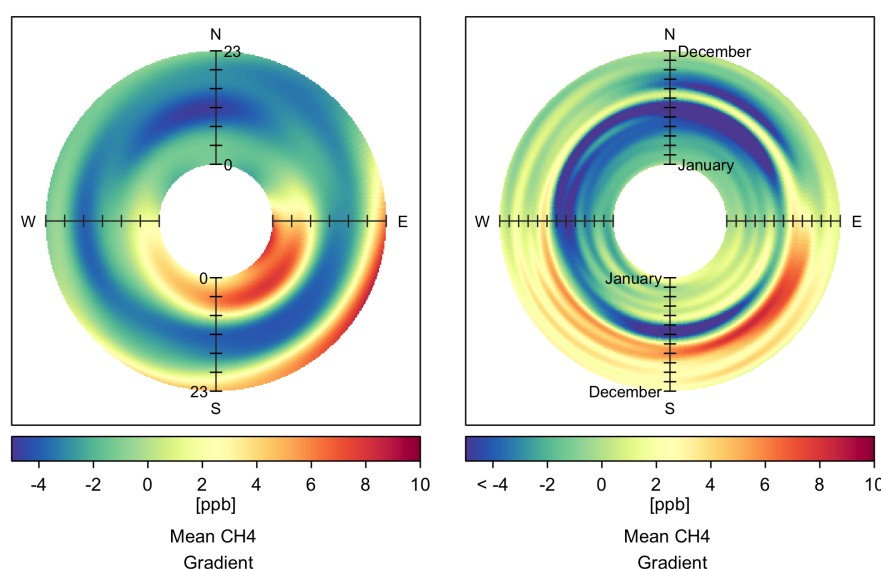

**Figure 7.** Mean CH$_4$ gradient for each bin of wind direction and time of day (left panel) or month of the year (right panel). Wind direction measured at 81 m and CH$_4$ mixing ratios at 79 m. Plot produced in R with the Openair package (Carslaw and Ropkins, 2012)

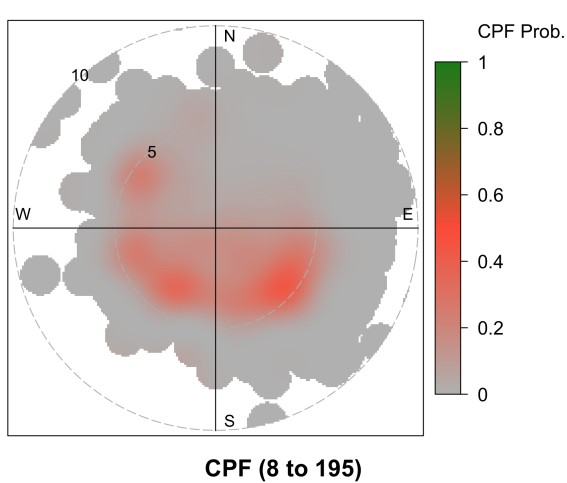

**CPF (8 to 195)**

**Figure 8.** Polar plot showing the conditional probability function (CPF) of the gradients above 8 ppb (88th to 100th percentiles, shown in the bottom of the graph as 8 to 195). The radial axis shows wind speed intervals and the colors the probability of having gradient above 88th percentile. This is shown for each bin composed by wind direction and wind speed at 81 m. Plot produced in R with the openair package (Carslaw and Ropkins, 2012)

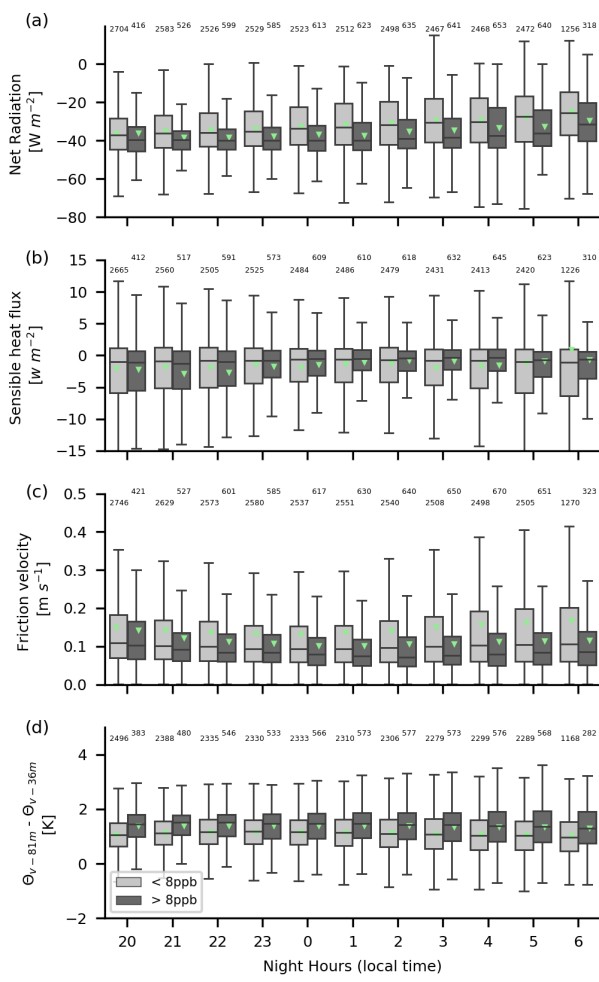

**Figure 9.** Box-and-whisker plots of net radiation (a), sensible heat flux (b), friction velocity (c), and virtual potential temperature difference (d) for nighttime hours. At each hour the gray colors indicate data points that correspond to either a gradient above 8 ppb or below 8 ppb. The means are indicated by the green triangles and the number of data points for each box-and-whisker plot is shown on the top of each panel. Note that the sensible heat flux and the friction velocity were measured at 81 m and net radiation at 75 m. Note that the air inlet for $CH_4$ mole ratios is at 79 m.

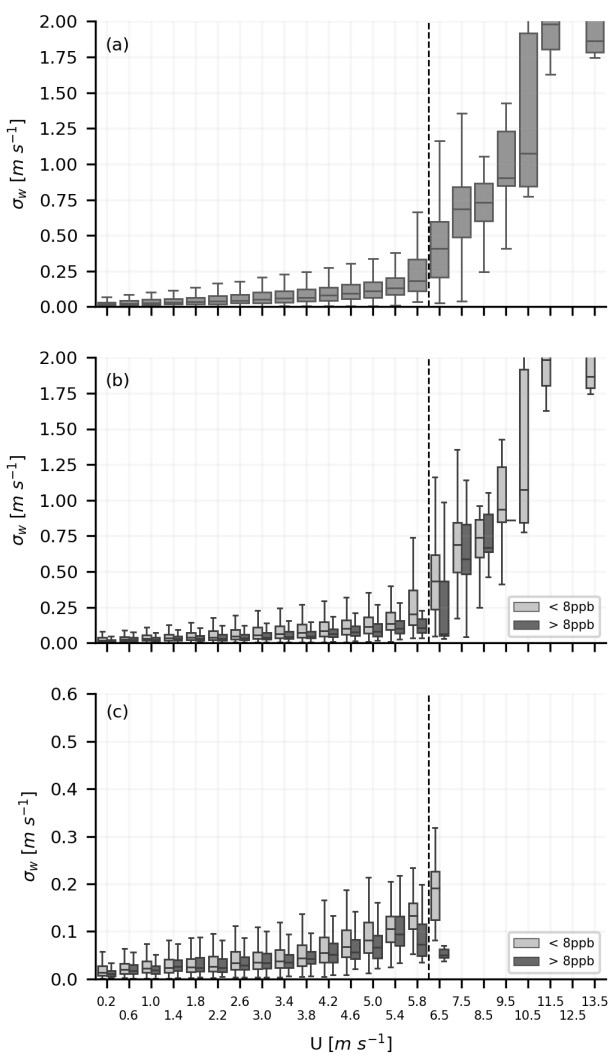

**Figure 10.** Standard deviation ($\sigma_w$) of the vertical velocity, plotted against the mean wind speed, for 1-minute averaging time. In the top panel (a), the variability of $\sigma_w$ for each wind speed bin is shown for all data points, with no classification based on $CH_4$ gradients. On (b), the same as (a) is shown, but the gradients below and above 8 ppb are separated. Note that our $CH_4$ mixing ratio measurements are at 15-minute resolution, therefore we assume the same value for every 15-minute window so we can associate the 1-minute $\sigma_w$ and U with $CH_4$ mixing ratios. In the bottom panel (c), we excluded the 15-minute time periods where the wind speed varied by more than 0.5 m s$^{-1}$. Note the difference in y-axis for (c). The wind speed bins are shown every 0.4 m s$^{-1}$ until 5.8 m s$^{-1}$, after the spacing is 1 m s$^{-1}$. This gradual increase to coarser wind speed bins is done due to sparser data at higher wind speeds. High-frequency measurements cover six months of 2014: March, April, May, July, August and September.





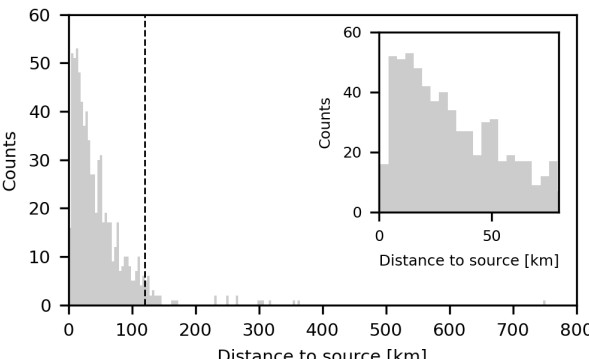

**Figure 11.** Distribution of the cumulative distance the wind at 81 m travelled from the potential source. The distance was calculated by time-integrating the wind speed at 81 m from 20:00 (beginning of the night) until the first occurrence of a positive gradient (> 8 ppb). The vertical dashed line shows the distance to the Amazon River in the southeast direction. The inset on the top right is a zoomed in view showing the x-axis until 80 km.

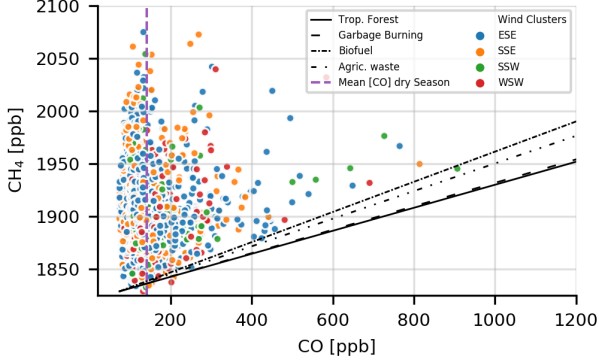

**Figure 12.** $CH_4$ as a function of carbon monoxide (CO) together with the slopes of the expected emission ratios (ER) for different types of fires based on the updated assessment of Andreae (2019) and classified into the dominant wind direction clusters for positive gradients (southern quadrants: from 90 to 270 degrees). The data points were filtered to select only nighttime measurements, and the CO data were further filtered to select the same times for which the above-8-ppb class was seen for our $CH_4$ mixing ratio measurements. The data cover the time period between June 2013 and November 2018. All measurements were performed at 79 m.





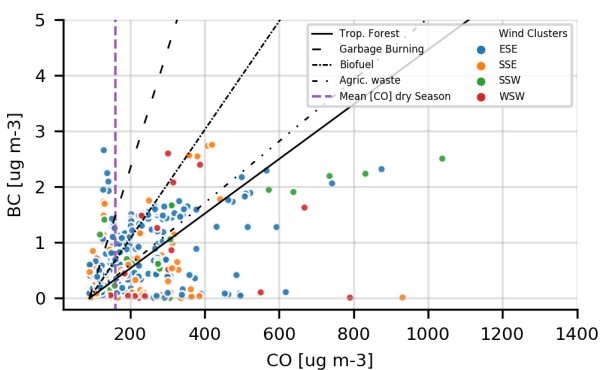

**Figure 13.** Black carbon as a function of carbon monoxide (CO) together with the slopes of the expected emission ratios (ER) for different types of fires based on the updated assessment of Andreae (2019) and classified into the dominant wind direction clusters for positive gradients (southern quadrants: from 90 to 270 degrees). The data points were filtered to select only nighttime measurements, and the BC data were further filtered to select the same times for which the above-8-ppb class was seen for our $CH_4$ mixing ratio measurements. Note that the time period of these data is from 2014 to 2015. The CO measurements are at 79 m and the BC are at 60 m



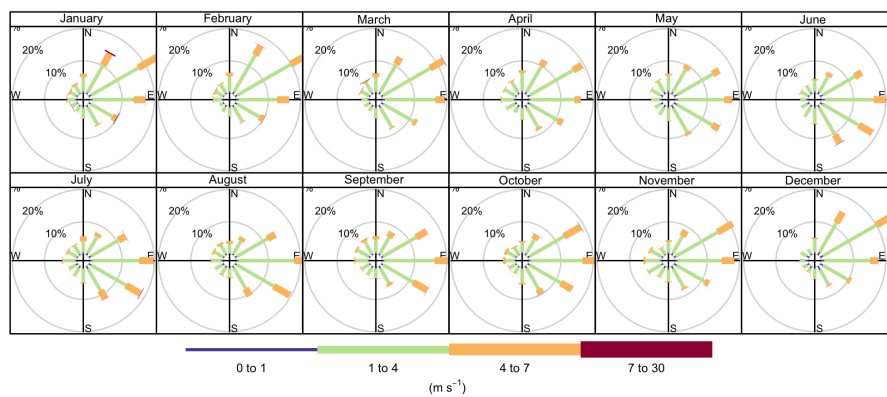

**Figure A1.** Monthly averaged wind rose plots at 81 m.

**Appendix A:  Mean monthly and mean hourly wind direction**

**Appendix B:  Radiation in wet and dry seasons at ATTO**

**Appendix C:  Dead stands of flooded forest**





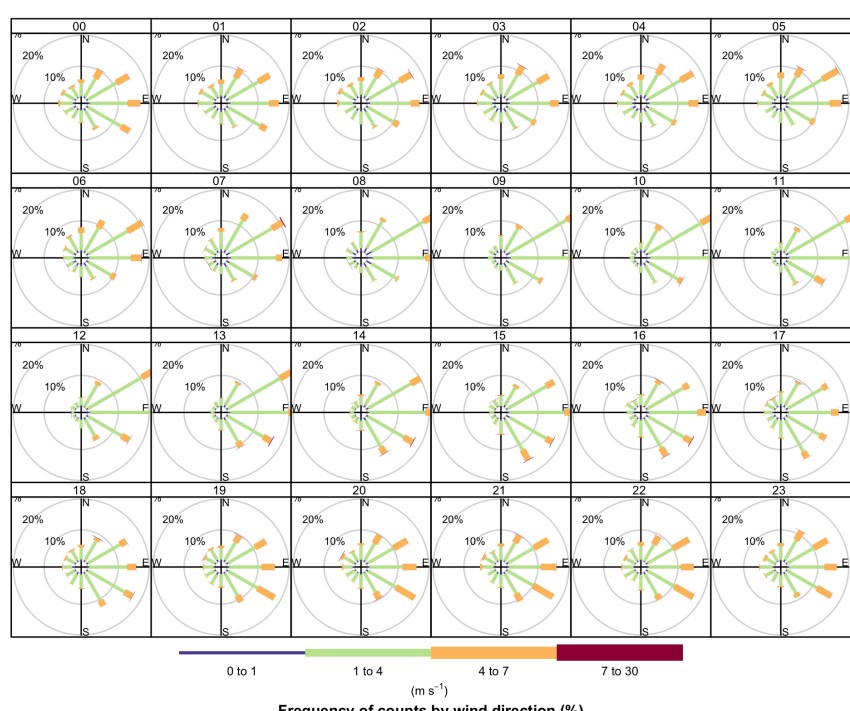

**Figure A2.** Hourly mean wind rose plots for each hour of the day. Averaged over all measurement period at 81m.

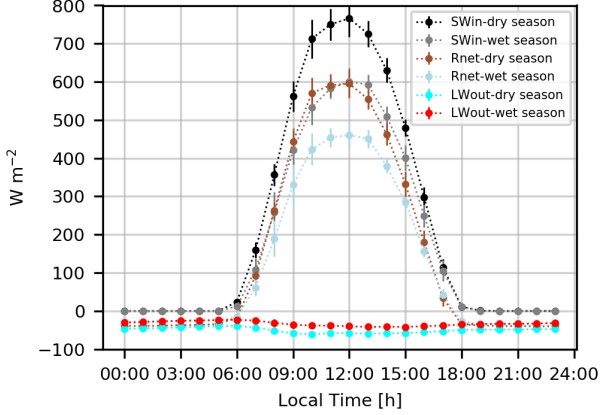

**Figure B1.** Net, short wave incoming, and long wave outgoing radiation for dry and wet seasons at ATTO.,



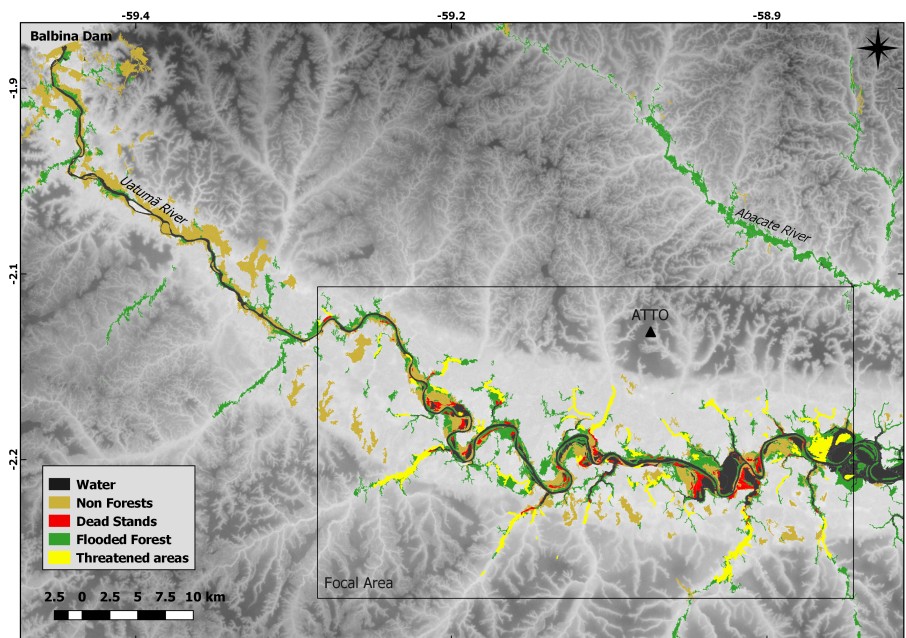

**Figure C1.** Map showing the dead stands and potentially threatened areas due to the tree mortality caused by the Balbina Dam. This map was modified from Science of The Total Environment, Vol 659, Authors: Angélica Faria de Resende, Jochen Schöngart, Annia Susin Streher, Jefferson Ferreira-Ferreira, Maria Teresa Fernandez Piedade, and Thiago Sanna Freire Silva, Massive tree mortality from flood pulse disturbances in Amazonian floodplain forests: The collateral effects of hydropower production, Pages 587-598, Copyright (2019), with permission from Elsevier.