# Peer review of "Understanding nighttime methane signals at the Amazon Tall Tower Observatory (ATTO)"

_Atmospheric Chemistry and Physics, 2019_

## Referee Comment (RC2) · Anonymous Referee #2 · 19 Feb 2020

Review of the manuscript by Botía et al

Title: Understanding nighttime methane signals at the Amazon Tall Tower Observatory (ATTO).

The study presents several years of CH4 mixing ratio profile measurements in the ATTO station, located in the Amazonian upland forest. The focus is on nighttime dynamics related to events when CH4 mixing ratio at 79 m inlet is higher than the one measured inside the canopy at 4 m. The dataset, spanning from June 2013 to November 2018, is unique, however the framework analysis and the structure of results/discussion could be improved. I can recommend the final publication in ACP after the following comments are properly addressed:

1) I suggest to re-structured the Results chapter. At moment, the different sub-sections include both results and discussion, and in some cases even methodology. Moreover, they are too lengthy and sometimes the reader loss the story line. So, I suggest to separate text related to results and discussion in two different sections. The chapter 3.3 and related sub-sections are mainly Discussions and not really presenting results. Finally, I see the needs to reduce the length of the text, as also suggested by the other Reviewer.

2) Most of the events associated with positive $CH_4$ gradient are associated with strong thermal inversions, low vertical mixing and decoupled regimes. I believe the decoupling happen somewhere between the canopy layer and the 79 m height. Could the authors see any systematic wind direction difference between the two layers? This would be a clear evidence supporting the idea of decoupling layer (see Alekseychik et al (2013) for an example of such analysis related to a boreal forest in Finland).

3) The authors suggest the nighttime $CH_4$ enhancement above the canopy (at 79m) are advected from the Uatuma river by horizontal non-turbulent motion. Assuming this is true, do the authors see also an increase of $CO_2$ mixing ratio at 79m? In fact it is evident that rivers (even in Amazonia region) are also a strong sources of $CO_2$ (authors can easily find few recent papers in literature related to this topic). However, for $CO_2$ the gradient would be still negative due to strong local sources related to nighttime soil respiration.

Minor comments:

- P2L21 and L23. $CH_4$ mixing ratio?

- P2L34. "….accumulation of $CH_4$ above the canopy…"

- P4L25-26. I would rephrase as "….wind speed profiles for specific nights, using additional data….".

- P5L26 – P6L2. This text can go under Discussion.

- P10L4-8. This text can go under Methodology, as it is related to how the data were analysed.

- Chapter 3.2.2 and figure 9. I would move this text and figure 9 to Appendix, or even removed from the manuscript. I am not sure about the meaning of friction velocity as surface layer scaling parameter in case of decoupling and/or very shallow NBL. I would rather suggest to look at some stability parameters, like bulk Richardson number (even calculated for different layers) or $z/L$, which would include both the mechanical forcing related to wind shear as well as the effect of buoyancy, which could act as suppression (stable conditions) or production (unstable conditions) mechanism for turbulent mixing.

- P12L9. Do you mean above-8-ppb $CH_4$ gradients?

- P12L9. How H is calculated? Have you corrected the kinematic heat flux $<w'T'>$ for the effect of $H_2O$ fluctuations? And if yes, how did you measure the $H_2O$ fluctuations? Was a Licor installed at 81m?

- P12L12. The heat fluxes close to zero may even indicate near-neutral atmospheric conditions, and it depends also on the relative magnitude of the wind shear forcing. See my comments above.

- P13L8-14. Can this behavior of the friction velocity profile (indicating actually a momentum flux divergence) be checked and analysed for the present dataset?

- P14L16. Please explain the difference between regimes 1 and 2.

- P14L25-26. I was thinking if this drop in the sigma_w the authors see for relatively moderate wind speed, could be related to the fact that 1 min average values are used for such analysis, which may filter out contribution of low frequency (e.g. submeso motions) to the std values.

- P15L4-11. This part looks more like methods, and not really results. Please define the Obukhov length L somewhere in the text.

- P15L11-15. How the authors can explain nighttime unstable conditions above the canopy? I have seen some other studies reporting unstable conditions within closed canopy (even in Amazonia), but not above it, as one would expected large cooling on the top of the canopy.

- P19L34. "....in this plot differs from that...."

- Fig 4. Why nighttime is defined including (probably for some seasons) sunset periods, but excluding sunrise hours? I understand that these atmospheric boundary layer transition periods are complicated, but they are also interesting. And are these criteria for separating nighttime and daytime holding for all seasons?

References:

Alekseychik, P., Mammarella, I., Launiainen, S., Rannik, Ü., Vesala, T., 2013: Evolution of the nocturnal decoupled layer in the pine forest canopy, Agricultural and Forest Meteorology, 174-175, 15-27.

---

## Author Comment (AC1) · 8 Apr 2020

Comment from Referee 1 - Received on January 15/2020

The format in which the response is addressed is:

1. Black text shows comment of referee. Comments are numerated as RC1.1, for the comment 1 of referee 1. The page and line referred to by the referee are also shown.

2. Blue text shows the author's response and it has the same logic for numeration (e.g. AR1.1).

3. Red text shows the changes to the manuscript.

4. *Italics, underline and red text shows citations from the manuscript*.

**General Comment**

[**RC1.1**] This is a generally well-written manuscript about understanding the reason why methane concentrations are occasionally strongly positive at nighttime in the Amazon forest. The authors describe that the positive gradients occur when the atmosphere is stable, with wind speed and direction in a certain range. They eliminate local sources, forest fires as potential explanations and conclude that the most probable explanation is remote sources from wet forest areas, where the methane is brought into the air by meso-scale type atmospheric circulations. As a plain observations driven analysis, this is a welcome and original scientific contribution. At the end the authors suggest that further research by modelling could strengthen the conclusions, and I agree, although the modelling will be challenging.

The manuscript is clearly written, most arguments are sound. I have two concerns, however. The first is: the paper is overly long. The results section is a mixture of results and discussion and even hypotheses. It takes a long time to read and this will scare off potentially interested readers. I suggest a general reduction in the order of 35% for all sections, which I think is feasible. My second concern is that there are a lot of hypotheses (as indicated by 'could', 'probably', 'potential', etc.). These hypotheses are mixed with real results. I suggest a better separation of the text in 'Results' into a section containing data-driven evidence and a section containing discussion (including hypotheses). In this way it will be easier for the audience to know what the data-analysis contribution of the authors is, and which part is still open for interpretation, further experiments, modelling, etc.. I think such a separation is more conform the journal's standard too. After these two concerns are addressed, I would advise publication of the manuscript in Atmos-Phys-Chem.

[**AR1.1**] We appreciate the referee's comments and believe they will improve the manuscript significantly. As the comment about the length of the paper was also made by referee 2, we include numerous changes in the manuscript that address this issue and reduce the two-column manuscript length (including figures) by three pages, without compromising the quality of our research. In

addition, the Results and Discussion section was separated, taking into account the potential confusion that the reader might encounter regarding our contribution and what is left for interpretation. We refer the referee and the editor to the revised version of the manuscript to see these changes. Furthermore, Figure 12 has been updated as more Black Carbon data became available in the meantime. The new time series extends from June 2013 to May 2018. This information has been updated in the methodology section accordingly.

**Specific Comments**

**[RC1.2]-P2,l31**: 'Vertical CH4 profiles . . . to decrease': the concentrations decrease or the profile has a negative gradient, but a decreasing profile is not semantically correct.

    **[AR1.2]** The sentence was changed to: Vertical $CH_4$ mixing ratios inside the canopy were found to decrease with height.

**[RC1.3]-P4,l28**: Rototronic to Rotronic.

    **[AR1.3]** The change was included.

**[RC1.4]-Section 2.5**: include units in the comparisons (e.g. $CH4_{grad} < 0$ ppb).

    **[AR1.4]** A new line was added to clarify the units in this section: We indicate here that the units of the $CH_{4_{grad}}$ and the comparisons are in parts per billion (ppb).

**[RC1.5]-Section 3, all subsections**: the text is too long, there is a lot of speculation, as a consequence the readers doesn't know what is fact and what is thinking.

    **[AR1.5]** This was addressed in the revised manuscript, we kindly refer the editor and the referee to this new version. Amongst the modifications we have included, the most important are: 1. Two new methods sections in which the 1-minute and 30-minute analysis' methodologies are briefly explained (see new sections 2.6 and 2.7), method-like parts were removed from Results. 2. The previous section 3.5 was moved to the discussion (see new sections 4.1 and 4.2). By doing this, most of our suggestions based on our results, are moved to the discussion. 3. We have shortened the Section 3 considerably. We believe these changes improve significantly the manuscript.

**[RC1.6]-P7,l12**: 'This directly affects. . .': this is not true per se. It depends on the gradient and the source. If the free atmospheric concentrations are higher than the PBL concentrations (positive gradient), more mixing will lead to higher concentrations near the ground, not lower.

    **[AR1.6]** This is completely true, we slightly modified the text to clarify this for our context. The new line is as follows: Under this case with higher mixing ratios close to the surface and lower free tropospheric mixing ratios (as shown for dry and wet seasons by Beck et al. (2012)), a deeper boundary layer directly affects daytime $CH_4$ mixing ratios because $CH_4$ enhancements near the surface will be diluted in a larger volume. This dilution effect does not happen at the 4 m inlet, because the within-canopy air volume remains the same throughout the seasons.

**[RC1.7]-P7,l21**: I don't think 4 m above the soil would be too high to measure the contribution from soil CH4 emission at night, particularly because many forest atmospheres are slightly unstable

at night because the canopy top is cooling faster than the soil.

[**AR1.7**] We agree with the referee here, for nighttime conditions this might be the case. However, for daytime conditions the 4 m inlet is high for attributing soil influence since the flow within the canopy is very often erratic during daytime with neutral to stable conditions. Reviewing the manuscript, we realized that the sentence was not intended for nighttime conditions within the canopy. There was a misunderstanding since it follows our reference to the largest negative gradient during nighttime. During the re-arrangement of the manuscript, in order to shorten it, we have taken out the following line: In contrast, the largest negative nighttime gradient measured is -236 ppb, occurring in October of 2015. This will also avoid any further confusion.

[**RC1.8**]**-P7,l30**: you derive a WFPS of 57%, which is interesting information. But on the basis of what do you deduct that CH4 production is enhanced? Could you show evidence of the variation of WFPS and from which level is methane production enhanced?

[**AR1.8**] Our reasoning here was based on Verchot et al. (2000), who found methane production hot spots (i.e., positive fluxes) during the wet season. They associated such positive fluxes with high methane concentrations in the soil profile. See specifically Figure 3 in Verchot et al. (2000), in which methane concentrations in what they call "Primary Forest", show spikes in the soil profile before 1 m and at near 5 m depth. Within the mechanisms that they attribute such methane production are: 1. Termites and 2. Enhanced anaerobic micro-sites due to an oxygen depletion in the soil profile, resulting from high rates of respiration. They indicate that most of the positive methane fluxes were observed at a WFPS above 60%. The level to which we attribute the WFPS is 60 cm depth, which is the depth that presents the maximum soil moisture in our profile. In our calculation we selected $0.35$ $m^3$ $m^{-3}$ for the volumetric soil moisture because this is the mean of our entire record at this depth, but this value often is higher during the wet season, yielding WFPS above 60%. On Fig. 1 the variation of WFPS at all depths with available soil moisture measurements is shown. Furthermore, taking into account the comments of the referee we have included the following lines in the manuscript to clarify our line of thought:

... we can calculate the water-filled pore space (WFPS) for the depth (60 cm) of maximum soil moisture content, at which we believed $CH_4$ could be produced. To be conservative we take the mean soil moisture value for the entire record at 60 cm, $0.35$ $m^3$ $m^{-3}$.

... This results in a WFPS of 57%, which is likely to enhance the abundance of anaerobic micro-sites where methanogenic bacteria can be activated. At values above 60% Verchot et al. (2000) found positive $CH_4$ fluxes, at the ATTO site values above 60% are often seen during the wet season.

[Figure]

Figure 1: Monthly time series of water-filled pore space at different depths. Measurements performed on *terra-firme* forest at the ATTO site.

**[RC1.9]-P8, section 3.1.1**: I wonder if it would be useful to give these examples earlier in the paper. I would have appreciated it.

[**AR1.9**] Considering the referee's comment here, we have decided to move this section to the beginning of section 3. This change, should improve the readability of the manuscript considerably.

**[RC1.10]-P8,section 3.1.1, first paragraph**: this is a verbal description of what can be easily seen in the figure. Please shorten and only highlight the aspects of importance.

[**AR1.10**] We agree with the referee's comment and decided to omit this short paragraph and start the section describing the results directly. This also contributes with shortening the manuscript.

**[RC1.11]-P8,l28:** ..... the profile decreases. The gradient decreases or the temperature decreases, but not the profile itself.

[**AR1.11**] Thank you very much for the comment. In the process of shortening the manuscript we have taken this line out of the text.

**[RC1.12]-P16,l28**: At this point I was wondering how you imagine the CH4 to reach the higher levels. It would make sense to tell the reader that section 3.3.2 is dedicated to explaining this transport mechanism.

[**AR1.12**] We have added a sentence at the end of the paragraph stating this. See the revised manuscript in particular, P13,l9-10.

**References:**

[revised manuscript text omitted]

30  an IRGA (LI-7500A, LI-COR Inc., USA). Air temperature is measured at 10 heights: 81, 73, 55, 40, 36, 26, 12, 4, 1.5 and 0.4 m (a.g.l) with a Termohygrometer (C215,  Rotronic Measurement Solutions, UK.). Net radiation is measured with a Net radiometer (NR-Lite2, Kipp and Zonen, Netherlands) at 75 m (a.g.l). For precipitation data, a rain gauge (TB4,

Hydrological Services Pty. Ltd., Australia) is installed at 81 m, and for soil moisture a water content reflectometer (CS615, Campbell Scientific Inc., USA) provides data for the depths: 0.1, 0.2, 0.3 0.4, 0.6 and 1 m.

**2.4 Time period of data used**

In the present study we have used $CH_4$ mixing ratio and meteorological data at different time resolutions. When mentioning meteorological data, we refer to the variables described in 2.3. To provide more clarity we specify what type of data were used in each section. In Sect. 3.2, 3.3.1 and 3.3.2 we used $CH_4$ mixing ratios and meteorological variables at 30-minute resolution. The $CH_4$ mixing ratio record covers the period from June 2013 to November 2018, which enabled us to study the diurnal and seasonal variability within this period. In Sect. 3.3.3, we used high-frequency (10 Hz) meteorological data, in particular all wind components (u, v and w), in order to associate turbulence regimes with $CH_4$ mixing ratios at 15-minute resolution. More on the assumptions to link high-frequency wind data with 15-minute mixing ratios is given below. In Sect. 3.4, we use 30-minute averages of $CH_4$, CO and Black Carbon (BC) to assess the influence of biomass burning emissions in our $CH_4$ signals. The raw BC mass concentrations at 1-minute time resolution were obtained using a Multi-angle Absorption Photometer (MAAP, model 5012, Thermo Fisher Scientific, Waltham, USA), as described in Saturno et al. (2018). The instrument measures the absorption coefficient of aerosol particles deposited on a filter, which is converted to BC mass concentration by assuming a mass absorption cross section of 6.6 $m^2$ $g^{-1}$. In Table 1, we provide a list of the data used in each section, specifying the time resolution and the period of time used.

**2.5 $CH_4$ gradient definition**

A $CH_4$ gradient is defined as $CH_{4_{grad}} = CH_{4_{79m}} - CH_{4_{4m}}$. We indicate that the units of the $CH_{4_{grad}}$ and the comparisons here are in parts per billion (ppb). We refer to a positive gradient when $CH_{4_{grad}} > 0$ ppb, or to a negative gradient when $CH_{4_{grad}} < 0$ ppb. Note that positive gradients are related to higher $CH_4$ mixing ratios at 79 m than at 4 m, while negative gradients to higher mixing ratios at 4 m. Throughout this paper we also use a 8 ppb threshold for classifying positive gradients and negative gradients. In Sect 3.2 we use three classes. The first one refers to very strong positive gradients ($CH_{4_{grad}} > 8$ ppb or above-8-ppb class); the second one to gradients in between -8 ppb and 8 ppb ($-8 < CH_{4_{grad}} < 8$ ppb); the third one to very strong negative gradients ($CH_{4_{grad}} < -8$ ppb). In Sect. 3.3, we have limited our analysis to two classes, $CH_{4_{grad}} > 8$ ppb, and $CH_{4_{grad}} < 8$ ppb (below-8-ppb class). The motivation to use 8 ppb as the threshold value is to leave out small mixing ratio variations and select very strong events. The $\pm$ 8 ppb threshold is conservative and filters for strong gradients, if we consider that the annual global increase in atmospheric $CH_4$ during the last three years was 7.06, 6.95 and 10.77 ppb $yr^{-1}$ for 2016, 2017 and 2018 respectively (Dlugokencky and NOAA). It is always stated in the text which of these classes is being considered.

**2.6   Analysis of 30-minute averages**

 In Sect. 3.2, the  30-min averages of CH4 mixing ratios were grouped into daytime and nighttime and further classified into the three classes as described in Sect. 2.5. In Sect. 3.3.1 strong CH$_4$ positive gradients were associated with wind direction 
[revised manuscript text omitted]
, 2018).  Under this case with higher mixing ratios close to the surface and lower free tropospheric mixing ratios (as shown for dry and wet seasons by Beck et al. (2012)), a deeper boundary layer directly affects daytime $CH_4$ mixing ratios because $CH_4$ enhancements near the surface will be diluted in a larger volume. This dilution effect does not happen at the 4 m inlet, because the within-canopy air volume remains the same throughout the seasons. This boundary layer effect together with higher $CH_4$ mixing ratios at 4 m compared to 79 m during the dry season yield a lower dry season daytime mean minimum of -5.0 ppb, whereas the mean minimum during the wet season is -2.2 ppb.

Another possibility that might contribute to this seasonal difference is local production of $CH_4$ during the wet season. Though we lack long-term $CH_4$ flux measurements at the site, we can infer a potential local source during the wet season considering that the mean monthly gradient during daytime hours of the wet season is always negative (not shown), meaning that the $CH_4$ mixing ratio at 4 m is higher than at 79 m. Although outside of the scope of this study, strong negative gradients are more common during daytime, reaching differences as large as -455 ppb, measured in May of 2014.  The 4 m inlet is too high above the soil to  directly associate this signal with the soil below, but is well within the canopy indicating that the source must be local possibly within a horizontal distance of few hundred meters. The event in May 2014, coincided with a strong signal measured for carbon monoxide (CO) with the same timing, but not for carbon dioxide ($CO_2$), which suggests a source not related to combustion.

The mechanism producing this strong $CH_4$ signal within the canopy is currently under investigation, yet here we discuss what the potential sources could be. Our first thought  is that the soil on the plateau is producing $CH_4$ episodically. Given some additional parameters, we can calculate the water-filled pore space (WFPS) for the depth (60 cm) of maximum soil moisture content, at which we believe $CH_4$ could be produced. To be conservative we take the mean soil moisture value for the entire record at 60 cm, 0.35 $m^3$ $m^{-3}$.  According to Andreae et al. (2015) 85% of the soil in the plateau is clay,  thus we use a soil particle density of

2.86 g cm$^{-3}$ (Schjønning et al., 2017)). Also from Andreae et al. (2015), we use a bulk density of 1.1 g cm$^{-3}$. This results in a WFPS of 57% , which is likely to enhance the abundance of anaerobic micro-sites where  methanogenic bacteria can be activated. At values above 60% Verchot et al. (2000) found positive CH$_4$ fluxes, at the ATTO site values above 60% are often seen during the wet season. 
[revised manuscript text omitted]

30  by the potential temperature gradient ($\frac{d\theta}{dz}$) between 81 m and 36 m. We found that 99.95% of the

 positive gradients occur with positive values of $\frac{d\theta}{dz}$. As a result, one can infer the absence of vertical  motions at the tower location. We can only attribute this lack of vertical mixing above the canopy to the tower location. Therefore, we have to separate the NBL conditions at the tower and at the potential source location. Unfortunately, we only have measurements at the tower and it is not realistic to measure at all possible source locations. Thus,  the transport mechanisms can be divided into 1. those responsible for vertical transport of $CH_4$ at the source location and 2. those responsible for the horizontal advection bringing the $CH_4$ signals to the tower. ~~Nocturnal vertical exchange, the mechanisms referred to in 1., can be driven by intermittent turbulence (Acevedo et al., 2006; Oliveira et al., 2018), gravity waves (Fitzjarrald and Moore, 1990), katabatic or drainage flows (Goulden et al., 2006; Tóta et al., 2008; Araújo et al., 2008; Tóta et al., 20 and nocturnal land-river breezes (De Oliveira and Fitzjarrald, 1994; Sun et al., 1998). For the horizontal transport of~~ More on the vertical and horizontal transport mechanisms is discussed in Sect. 4.2.

**3.4 Rejecting biomass burning and the Amazon river as potential sources**

[revised manuscript text omitted]

---

## Author Comment (AC2) · 8 Apr 2020

Comment from Referee 2 - Received on February 19/2020

The format in which the response is addressed is:

1. Black text shows comment of referee. Comments are numerated as RC2.1, for the comment 1 of referee 2. The page and line referred to by the referee are also shown.

2. Blue text shows the author's response and it has the same logic for numeration (e.g. AR2.1).

3. Red text shows the changes to the manuscript.

4. *Italics, underline and red text shows citations from the manuscript*.

**General Comments**

[**RC2.1**] I suggest to re-structured the Results chapter. At moment, the different sub-sections include both results and discussion, and in some cases even methodology. Moreover, they are too lengthy and sometimes the reader loss the story line. So, I suggest to separate text related to results and discussion in two different sections. The chapter 3.3 and related sub-sections are mainly Discussions and not really presenting results. Finally, I see the needs to reduce the length of the text, as also suggested by the other Reviewer.

[**AR2.1**] We appreciate the referee's comments and believe shortening the text will be beneficial for the manuscript. The separation of Results and Discussion was done and we refer the referee to the revised manuscript. Sections 3.3.1 and 3.3.2 were added as a part of the Discussion, yet **Section 3.3.3 Rejecting Biomass Burning Influence on positive $CH_4$ gradients** was left in the Results. This Section was renamed as **Rejecting biomass burning and Amazon river as sources of the positive $CH_4$ gradients** because the part in which we discard the Amazon river as source of the positive $CH_4$ gradients was added. The motivation for this is that this part is based on analysis of our data. Furthermore, and as we stated to the referee 1, Figure 12 has been updated as more Black Carbon data became available in the meantime. The new time series extends from June 2013 to May 2018. This information has been updated in the methodology section accordingly.

[**RC2.2**] Most of the events associated with positive CH4 gradient are associated with strong thermal inversions, low vertical mixing and decoupled regimes. I believe the decoupling happen somewhere between the canopy layer and the 79 m height. Could the authors see any systematic wind direction difference between the two layers? This would be a clear evidence supporting the idea of decoupling layer (see Alekseychik et al., 2013) for an example of such analysis related to a boreal forest in Finland).

[**AR2.2**] Thanks for your suggestion, we have analyzed our dataset from this perspective. First, is worth recalling that our $CH_4$ inlet is at 79 m and the closest sonic anemometer is at 81 m, we have stated this in P11-l24: *The heights of the highest sonic anemometer and the highest air inlet for $CH_4$*

*mixing ratio measurements differ by two meters, with the former at 81 m and the latter at 79 m. We assume that the effect of the two meters can be neglected and thus interpret all the 81 m data as valid for 79 m.*. We found a systematic difference in wind direction under $CH_4$ enhancement events but also, when they do not occur (see Fig. 1). This plot was based on Figure 5 in Alekseychik et al. (2013), the reference suggested by referee 2. This indicates that nighttime conditions at ATTO are often decoupled showing wind direction differences between levels. Decoupling conditions of the air within and above the canopy during nighttime were first described by Shuttleworth (1985) and at the ATTO site were mentioned by Andreae et al. (2015). Here we show that the decoupling holds from 19 m to 26 m but at 42 m the wind direction has a similar distribution to that at the upper level. It is important to note here that under positive $CH_4$ gradients in addition to the wind direction difference, there are marked differences in net radiation, sensible heat flux, friction velocity and virtual potential gradient as depicted by Figure 9 of the manuscript. As this analysis does not influence our results we have decided to not include this plot in the revised manuscript.

[Figure]

Figure 1: Wind direction distribution at 81 and 19 m levels (left column), at 81 and 26 m (middle column) and at 81 m and 42 m (right column). The upper row shows the below-8-ppb class, the middle row the between -8 and 8 ppb class and the lower row the above-8-ppb class. The vertical lines indicate the mean values for each distribution. See the manuscript for further details on our nighttime definition and the last comment in this document. Note that the distribution is limited to the availability of data at both levels, this is why the comparison between 81 m and 42 m levels differ from the others in the same row.

[**RC2.3**] The authors suggest the nighttime CH4 enhancement above the canopy (at 79m) are advected from the Uatumã river by horizontal non-turbulent motion. Assuming this is true, do the authors see also an increase of CO2 mixing ratio at 79m? In fact it is evident that rivers (even in Amazonia region) are also a strong sources of CO2 (authors can easily find few recent papers in literature related to this topic). However, for CO2 the gradient would be still negative due to strong local sources related to nighttime soil respiration.

[**AR2.3**] As the referee states, the $CO_2$ gradient will always be negative due to very strong nighttime respiration. Therefore, is difficult to identify a $CO_2$ increase with our gradient approach. We do not discard that an increase in $CO_2$ could happen, but the magnitude of such enhancement is very often masked by very strong $CO_2$ signals that are locally produced at lower levels. An example of this is shown on Fig. 2. At the time of $CH_4$ enhancement (at around 21:00) there is a very small

increase in $CO_2$ mixing ratios at 79 m and not seen at 24 m, this can be possibly attributed to an enhancement coming from the Uatumã river. However, later in the night a bottom-up strong local signal arrives at 79 m at about 03:00, masking the earlier effect. A short sentence taking into account this comment was added to the revised manuscript. See section 4.1.

[Figure]

Figure 2: Time series for a 24-hour interval (same as in Fig 6a in manuscript) for $CO_2$ and $CH_4$.

**Minor Comments**

[**RC2.4**]-**P2,l21 and l23**: CH4 mixing ratio?

[**AR2.4**] From this comment, we interpret that the referee is suggesting to define the mixing ratio measure earlier in the text. Since this is a common and well-known measure for concentrations of chemical species in the atmosphere, we believe is not necessary to define it.

[**RC2.5**]-**P2,l34**: "...accumulation of CH4 above the canopy..."

[**AR2.5**] We have corrected the sentence as suggested by the referee.

[**RC2.6**]-**P4,l25-26**: I would rephrase as "....wind speed profiles for specific nights, using additional data....".

[**AR2.6**] Thank you for the comment, we have modified the text accordingly.

[**RC2.7**]-**P5,l26 to P6,l2**: This text can go under Discussion.

[**AR2.7**] Yes, we agree with the comment and have included this part under Discussion.

[**RC2.8**]-**P10,l4-8**: This text can go under Methodology, as it is related to how the data were analysed.

[**AR2.8**] Thanks for the suggestion, the text has been added to the Methods section.

**[RC2.9]-Chapter 3.2.2 and Figure 9**: I would move this text and figure 9 to Appendix, or even removed from the manuscript. I am not sure about the meaning of friction velocity as surface layer scaling parameter in case of decoupling and/or very shallow NBL. I would rather suggest to look at some stability parameters, like bulk Richardson number (even calculated for different layers) or z/L, which would include both the mechanical forcing related to wind shear as well as the effect of buoyancy, which could act as suppression (stable conditions) or production (unstable conditions) mechanism for turbulent mixing.

[**AR2.9**] We understand the perspective of the referee 2, however we do not think Section 3.2.2 and Figure 9 should be left out of the manuscript nor moved to the Appendix. The main objective of this section was to show the effects of some parameters associated with atmospheric stability, such as sensible heat flux ($H$) and friction velocity ($u_*$), on the $CH_4$ positive gradients. The friction velocity was used here as a direct measure of mechanical turbulence and not as a surface layer scaling parameter. On Figure 9 of the manuscript was shown that the strong $CH_4$ gradients (above-8-pbb class) were associated with high negative sensible heat flux values and low values of friction velocity. Therefore, for situations with strong $CH_4$ gradients, it is expected that the Obuhkov length (L) will present smaller values than the other class (below-8-pbb class). In Fig. 3 the z/L values for the two $CH_4$ classes investigated are shown. As expected, the z/L values were generally higher for the above-8-pbb (since L is lower) than for the below-8-ppb class.

[Figure]

Figure 3: Nighttime evolution of z/L for the two classes analyzed in Section 3.2.2 above-8-pbb and below-8-ppb.

**[RC2.10]-P12,l9**: Do you mean above-8-ppb CH4 gradients?

[**AR2.10**] Yes. Thanks for pointing this out. The sentence was corrected for clarity.

**[RC2.11]-P12,l9**: How H is calculated? Have you corrected the kinematic heat flux w'T' for the effect of H2O fluctuations? And if yes, how did you measure the H2O fluctuations? Was a Licor installed at 81m?

[**AR2.11**] In this section we use 30-min averages that are processed in the EddyPro/Alteddy softwares. The correction for the effect of water fluctuations is performed with information from an IRGA (LI-7500A, LI-COR Inc., USA) located at 81 m. We included this in the Meteorological Instrumentation part as: At 81 m molar densities of $CO_2$ and $H_2O$ are measured with an IRGA (LI-7500A, LI-COR Inc., USA).

[**RC2.12**]-**P12,l12**: The heat fluxes close to zero may even indicate near-neutral atmospheric conditions, and it depends also on the relative magnitude of the wind shear forcing. See my comments above.

[**AR2.12**] We agree with the referee 2, sensible heat fluxes close to zero may also indicate near-neutral conditions. Thus, we interpret the referee's comment about the relative magnitude of wind shear forcing, as a suggestion for an additional argument that can explain the lower sensible heat flux values for the above-8-pbb class. However, as we show on Fig 8 of the manuscript, the probability of having $CH_4$ gradients above 8 ppb was higher for wind speeds within a range between 2 and 5 m s$^{-1}$ (explicitly stated in P13,l4), so for relatively high wind shear at moderate wind speeds (above 4 m s$^{-1}$) we could also have methane enhancement events. This is the reason why when presenting the results for sensible heat flux in this part of the manuscript, we limited the analysis to stability and the nocturnal boundary layer height.

[**RC2.13**]-**P13,l8-14**: Can this behavior of the friction velocity profile (indicating actually a momentum flux divergence) be checked and analysed for the present dataset?

[**AR2.13**] Although we don't have friction velocity data for as many levels as Acevedo et al. (2016) had, we have checked this behaviour at 81 m and 46 m for two temporal scales, 30-minute and 1-minute averages. Unfortunately, we only have 1-minute data at both heights for August in 2014, yet the 30-minute record extends from 2013 to 2018 for both heights. The 1-minute data requires to be requested with some time in advance and we are limited to what we have at the moment. We took 46 m as the near-surface level since the canopy is just below 40 m and averaged the friction velocity for different wind speed bins at 46 m. The results of this are shown in Fig. 4. In the decoupled regime described in Acevedo et al. (2016), friction velocity at higher levels is increasingly driven by submeso motions. At ATTO the contribution of submeso motions is less dominant (Acevedo et al., 2014), explaining why we see a negative gradient of friction velocity (Fig. 4) and also why it does not change signs as observed by Acevedo et al. (2016). Our final decision considering that we needed to shorten the manuscript and that this part is not fundamental for the main conclusions, we decided to take these lines [P13,l8-14] out of the revised manuscript.

[Figure]

Figure 4: Friction velocity at 81 and 46 m as a function of wind speed at 46 m. Plotted using 30-minute averages (left panel) and 1-minute averages (right panel).

[**RC2.14**]-**P14,l16**: Please explain the difference between regimes 1 and 2.

[**AR2.14**] The main characteristics of regimes 1 and 2 are explained three lines below, (i.e. P14,l19). However, in the revised version we have placed these lines earlier in the text.

[**RC2.15**]-**P14,l25-26**: I was thinking if this drop in the $\text{sigma}_w$ the authors see for relatively moderate wind speed, could be related to the fact that 1 min average values are used for such analysis, which may filter out contribution of low frequency (e.g. submeso motions) to the std values.

[**AR2.15**] The 1 min average values follows recent practice in stable boundary layer studies (Mahrt et al., 2013; Acevedo et al., 2016; Stiperski et al., 2019). Moreover, it has been also applied for producing what is commonly known as the "Hockey stick plot" (Sun et al., 2012, Acevedo et al., 2016, 2019 and Mortarini et al., 2019), Figure 10 of the manuscript. Furthermore, the contribution from submeso motions, if not filtered using a small averaging window, may produce unpredictable large-scale contributions to vertical fluxes, increasing the variability associated with most of the quantities (Mahrt et al., 2013; Stefanello et al., 2020). Therefore, we believe that for our purpose –analyzing the positive gradients from the hockey stick plot perspective– we had to filtered out submeso motions to purely target turbulent scales. Nevertheless, under very stable conditions the time-scale of turbulent motions can decrease (Oliveira et al., 2018), thus the 1-minute averages can still be affected by submeso motions. In Fig. 5 we have used a larger averaging window (5 minutes) and we still see the drop in $\text{sigma}_W$ and the same wind speed threshold.

[Figure]

Figure 5: Same as Fig 10 in manuscript but using 5-minute averages.

**[RC2.16]-P15,l4-11**: This part looks more like methods, and not really results. Please define the Obukhov length L somewhere in the text.

[**AR2.16**] We have moved these lines to a new section in Methods. As explained in the following comment we have decided to not use the Obukhov length, therefore no definition is needed.

**[RC2.17]-P15,l11-15**: How the authors can explain nighttime unstable conditions above the canopy? I have seen some other studies reporting unstable conditions within closed canopy (even in Amazonia), but not above it, as one would expected large cooling on the top of the canopy.

[**AR2.17**] The referee is correct when indicating that the expected nighttime conditions above the canopy are stable. However, several studies have already shown that under certain conditions the stable layer above the Amazon canopy may break down (e.g. Fitzjarrald and Moore, 1990; Zeri and Sá 2011; Oliveira et al., 2018). One of the first works to report nighttime unstable conditions above

the Amazon canopy was Fitzjarrald and Moore (1990). According to them, there are cases when net radiative heat losses are sharply reduced at night, most likely because of increasing cloud cover. With reduced net radiation, there is eventually much-reduced cooling at canopy top to maintain the stable potential temperature. Recently Oliveira et al. (2018) also have shown that intermittent turbulence events, which are quite common in the Amazon NBL (frequently caused by the deep convection) can cause vertical exchange of scalars and have an effect on the local stability.

Furthermore, our interpretation of the results shown in Table 2 of the manuscript is the following. As we stated earlier, under very stable conditions submeso motions can still be relevant even at an averaging window of one minute. Under these situations and with weak wind speeds there is no turbulence, therefore the sensible heat flux can still be influenced by non-turbulent motions that can lead to negative values of L (with positive values of w'T', see Fig. 6-lower panel). In addition, the "very unstable" cases are also influenced by very small values of friction velocity (Fig. 6-middle panel). For the "moderately unstable" cases we see that at very rare occasions the potential temperature gradient is negative (see Fig 6-top panel). This can be explained by the advection of cold air, which leads to unstable conditions of the nocturnal boundary layer.

Considering how L can be affected and how using a "very unstable" regime classification for a study dealing with the stable boundary layer can lead to confusion, we have decided to use a different approach. Using the $\frac{d\theta}{dz}$ between 81 and 36 m will without doubt provide a cleaner assessment of stability. This is different than what is shown in Fig. 9 of the manuscript, because here we are using 1-minute data and taking into account the height difference. When separating between negative and positive values for $\frac{d\theta}{dz}$ we found that 99.95% of the above-8-ppb gradients are above zero. We have taken out Table 2 and added the following lines: More evidence about the dominant regime of the NBL when the positive gradients occur is given by the potential temperature gradient ($\frac{d\theta}{dz}$) between 81 m and 36 m. We found that 99.95% of the positive gradients occur with positive values of $\frac{d\theta}{dz}$.

[Figure]

Figure 6: Potential temperature gradient violin plot showing the entire distribution of the data (top panel), friction velocity and w'T' box plots (middle and lower panel) for each of the stability classes defined in the manuscript based on Kruijt et al. (2000), Table 2 of the manuscript.

**[RC2.18]-P19,l34**: "....in this plot differs from that...."

    **[AR2.18]** Thanks for pointing this typo out. We have corrected it.

**[RC2.19]-Fig 4**: Why nighttime is defined including (probably for some seasons) sunset periods, but excluding sunrise hours? I understand that these atmospheric boundary layer transition periods are complicated, but they are also interesting. And are these criteria for separating nighttime and daytime holding for all seasons?

[**AR2.19**] We believe that the referee meant: Why nighttime is defined including (probably for some seasons) *sunrise* periods, but excluding *sunset* hours?. We have excluded sunset hours in the nighttime interval to avoid the afternoon transition period, in which the turbulent kinetic energy begins to decay rapidly within the boundary layer. This transition period influences the distribution and mixing of trace species, for example Vila-Guerau de Arellano et al. (2004) showed that during the afternoon transition chemical species emitted at the surface can rapidly reach the free troposphere. The onset of the positive gradient events studied here is, in most of the cases, late in the night as described in P6,l34-P7,l4, and are sustained until early morning. Therefore, by selecting this nighttime period we wanted to cover the full positive gradient episodes but without affecting early nighttime averages due to the afternoon transition. Thanks to the comment of the referee we realized that for Fig 4, we included 06:00 for both nighttime and daytime, using that data point in both populations. We have corrected this considering daytime hours from 07:00 to 18:00 and updated the figure. This subtle change did not lead to substantial differences in our results, only daytime percentages discussed in P6,l27-28, were slightly affected. In the revised manuscript these percentages are updated to the new values.

**2.5 $CH_4$ gradient definition**

 A $CH_4$ gradient  is defined as $CH_{4_{grad}} = CH_{4_{79m}} - CH_{4_{4m}}$. We indicate that the units of the $CH_{4_{grad}}$ and the comparisons here are in parts per billion (ppb). We refer to a positive gradient when $CH_{4_{grad}} > 0$ ppb, or to a negative gradient when $CH_{4_{grad}} < $  0 ppb. Note that positive gradients are related to higher $CH_4$ mixing ratios at 79 m than at 4 m, while negative gradients to higher mixing ratios at 4 m. Throughout this paper we also use a 8 ppb threshold for classifying positive gradients and negative gradients. In Sect 3.2 we use three classes. The first one refers to very strong positive gradients ($CH_{4_{grad}} > 8$ ppb or above-8-ppb class); the second one to gradients in between -8 ppb and 8 ppb ($-8 < CH_{4_{grad}} < 8$ ppb); the third one to very strong negative gradients ($CH_{4_{grad}} < -8$ ppb). In Sect. 3.3, we have limited our analysis to two classes, $CH_{4_{grad}} > 8$ ppb, and $CH_{4_{grad}} < 8$ ppb (below-8-ppb class). The motivation to use 8 ppb as the threshold value is to leave out small mixing ratio variations and select very strong events. The $\pm$ 8 ppb threshold is conservative and filters for strong gradients, if we consider that the annual global increase in atmospheric $CH_4$ during the last three years was 7.06, 6.95 and 10.77 ppb $yr^{-1}$ for 2016, 2017 and 2018 respectively (Dlugokencky and NOAA). It is always stated in the text which of these classes is being considered.

**2.6 Analysis of 30-minute averages**

**3**

**2.1**

 In Sect. 3.2, the  30-min averages of CH4 mixing ratios were grouped into daytime and nighttime and further classified into the three classes as described in Sect. 2.5. In Sect. 3.3.1 strong $CH_4$ positive gradients were associated with wind direction 
[revised manuscript text omitted]

15  deeper boundary layer directly affects daytime $CH_4$ mixing ratios  because $CH_4$  enhancements near the surface will be diluted in a larger volume. This dilution effect does not happen at the 4 m inlet, because the within-canopy air volume remains the same throughout the seasons. This boundary layer effect together with higher $CH_4$ mixing ratios at 4 m compared to 79 m during the dry season  yield a lower dry season daytime mean minimum of -5.0 ppb, whereas the mean minimum during the wet season is -2.2 ppb.

20  Another possibility that might contribute to this seasonal difference is local production of $CH_4$ during the wet season. Though we lack long-term $CH_4$ flux measurements at the site, we can infer a potential local source during the wet season considering that the mean monthly gradient during daytime hours of the wet season is always negative (not shown), meaning that the $CH_4$ mixing ratio at 4 m is higher than at 79 m. Although outside of the scope of this study, strong negative gradients are more common during daytime, reaching differences as large as -455 ppb, measured in May of 2014.

25   The 4 m inlet is too high above the soil to  directly associate this signal with the soil below, but is well within the canopy  indicating that the source must be local possibly within a horizontal distance of few hundred meters. The event in May 2014, coincided with a strong signal measured for carbon monoxide (CO) with the same timing, but not for carbon dioxide ($CO_2$), which suggests a source not related to combustion.

30  The mechanism producing this strong $CH_4$ signal within the canopy is currently under investigation, yet here we discuss what the potential sources could be. Our first thought  is that the soil on the plateau is producing $CH_4$ episodically. Given some additional parameters, we can calculate the water-filled pore space (WFPS) for the depth (60 cm) of maximum soil moisture content, at which we believe $CH_4$ could be produced. To be conservative we take the mean soil moisture value for the entire record at 60 cm, 0.35 $m^3$ $m^{-3}$.

35   According to Andreae et al. (2015) 85% of the soil in the plateau is clay,  thus we use a soil particle density of

2.86 g cm$^{-3}$ (Schjønning et al., 2017)). Also from Andreae et al. (2015), we use a bulk density of 1.1 g cm$^{-3}$. This results in a WFPS of 57% , which is likely to enhance the abundance of anaerobic micro-sites where  methanogenic bacteria can be activated. At values above 60% Verchot et al. (2000) found positive CH$_4$ fluxes, at the ATTO site values above 60% are often seen during the wet season. 
[revised manuscript text omitted]
  BC data  set spans from  June 2013 to  May 2018. The heights of CO and BC measurements are  79  and  60 m a.g.l., respectively.

[Figure]

**Figure 13.** Distribution of the cumulative distance the wind at 81 m travelled from the potential source. The distance was calculated by time-integrating the wind speed at 81 m from 20:00 (beginning of the night) until the first occurrence of a positive gradient (> 8 ppb). The vertical dashed line shows the distance to the Amazon River in the southeast direction. The inset on the top right is a zoomed in view showing the x-axis until 80 km.

[Figure]

**Figure A1.** Monthly averaged wind rose plots at 81 m.

**Appendix A:  Mean monthly and mean hourly wind direction**

**Appendix B:  Radiation in wet and dry seasons at ATTO**

**Appendix C:  Dead stands of flooded forest**

[Figure]

**Figure A2.** Hourly mean wind rose plots for each hour of the day. Averaged over all measurement period at 81m.

[Figure]

**Figure B1.** Net, short wave incoming, and long wave outgoing radiation for dry and wet seasons at ATTO.,

[Figure]

**Figure C1.** Map showing the dead stands and potentially threatened areas due to the tree mortality caused by the Balbina Dam. This map was modified from Science of The Total Environment, Vol 659, Authors: Angélica Faria de Resende, Jochen Schöngart, Annia Susin Streher, Jefferson Ferreira-Ferreira, Maria Teresa Fernandez Piedade, and Thiago Sanna Freire Silva, Massive tree mortality from flood pulse disturbances in Amazonian floodplain forests: The collateral effects of hydropower production, Pages 587-598, Copyright (2019), with permission from Elsevier.